# Environment as a limiting factor of the historical global spread of mungbean

Pei-Wen Ong[1], Ya-Ping Lin[2,3], Hung-Wei Chen[2], Cheng-Yu Lo[2], Marina Burlyaeva[4], Thomas Noble[5], Ramakrishnan Madhavan Nair[6], Roland Schafleitner[3], Margarita Vishnyakova[4], Eric Bishop-von-Wettberg[7,8], Maria Samsonova[8], Sergey Nuzhdin[9], Chau-Ti Ting[10], Cheng-Ruei Lee[1,2]*

[1]Institute of Plant Biology, National Taiwan University, Taipei, Taiwan; [2]Institute of Ecology and Evolutionary Biology, National Taiwan University, Taipei, Taiwan; [3]World Vegetable Center, Tainan, Taiwan; [4]N.I. Vavilov All-Russian Institute of Plant Genetic Resources (VIR), St. Petersburg, Russian Federation; [5]Department of Agriculture and Fisheries, Warwick, Australia; [6]World Vegetable Center, South and Central Asia, Patancheru, India; [7]Department of Plant and Soil Science and Gund Institute for the Environment, University of Vermont, Burlington, United States; [8]Department of Applied Mathematics, Peter the Great St. Petersburg Polytechnic University, Saint Petersburg, Russian Federation; [9]University of Southern California, Los Angeles, United States; [10]Department of Life Science, National Taiwan University, Taipei, Taiwan

*For correspondence:
chengrueilee@ntu.edu.tw

Competing interest: The authors declare that no competing interests exist.

**Abstract** While the domestication process has been investigated in many crops, the detailed route of cultivation range expansion and factors governing this process received relatively little attention. Here, using mungbean (*Vigna radiata* var. *radiata*) as a test case, we investigated the genomes of more than 1000 accessions to illustrate climatic adaptation's role in dictating the unique routes of cultivation range expansion. Despite the geographical proximity between South and Central Asia, genetic evidence suggests mungbean cultivation first spread from South Asia to Southeast, East and finally reached Central Asia. Combining evidence from demographic inference, climatic niche modeling, plant morphology, and records from ancient Chinese sources, we showed that the specific route was shaped by the unique combinations of climatic constraints and farmer practices across Asia, which imposed divergent selection favoring higher yield in the south but short-season and more drought-tolerant accessions in the north. Our results suggest that mungbean did not radiate from the domestication center as expected purely under human activity, but instead, the spread of mungbean cultivation is highly constrained by climatic adaptation, echoing the idea that human commensals are more difficult to spread through the south-north axis of continents.

## Editor's evaluation

This is an important interdisciplinary effort, with compelling genetic evidence, that informs on the spread of an important crop. The work will be of broad interest to those studying the domestication and dissemination of cultivated plants.

## Introduction

Domestication is a process that is cultivated by humans, leading to associated genetic and morphological changes. These changes may be intentional from human selection or unintentional as a result of adaptation to the environments of cultivation (*Fuller, 2007*). Later, the cultivated plants spread out

**eLife digest** Mungbean, also known as green gram, is an important crop plant in China, India, the Philippines and many other countries across Asia. Archaeological evidence suggests that humans first cultivated mungbeans from wild relatives in India over 4,000 years ago. However, it remains unclear how cultivation has spread to other countries and whether human activity alone dictated the route of the cultivated mungbean's expansion across Asia, or whether environmental factors, such as climate, also had an impact.

To understand how a species of plant has evolved, researchers may collect specimens from the wild or from cultivated areas. Each group of plants of the same species they collect in a given location at a single point in time is known collectively as an accession. Ong et al. used a combination of genome sequencing, computational modelling and plant biology approaches to study more than 1,000 accessions of cultivated mungbean and trace the route of the crop's expansion across Asia.

The data support the archaeological evidence that mungbean cultivation first spread from South Asia to Southeast Asia, then spread northwards to East Asia and afterwards to Central Asia. Computational modelling of local climates and the physical characteristics of different mungbean accessions suggest that the availability of water in the local area likely influenced the route. Specifically, accessions from arid Central Asia were better adapted to drought conditions than accessions from wetter South Asia. However, these drought adaptations decreased the yield of the plants, which may explain why the more drought tolerant accessions have not been widely grown in wetter parts of Asia.

This study shows that human activity has not solely dictated where mungbean has been cultivated. Instead, both human activity and the various adaptations accessions evolved in response to their local environments shaped the route the crop took across Asia. In the future these findings may help plant breeders to identify varieties of mungbean and other crops with drought tolerance and other potentially useful traits for agriculture.

from their initial geographical range (*Meyer and Purugganan, 2013*), and elucidating the factors affecting the range expansion of crops is another focus of active research (*Gutaker et al., 2020*). In the old world, during the process of 'prehistoric food globalization' (*Jones et al., 2011*), crops originated from distinct regions were transported and grown in Eurasia. Archeological evidence has shown that such 'trans-Eurasian exchange' had happened by 1500 BC (*Liu et al., 2019*), and the proposed spread routes from archeological studies were supported by modern genetic evidence especially in rice (*Gutaker et al., 2020*) and barley (*Lister et al., 2018*). Interestingly, the spread may accompany genetic changes for the adaptation to novel environments. For example, in barley, variations in the gene *Photoperiod-H1* (*Ppd-H1*) resulting in the non-responsiveness to longer daylengths were likely associated with the historical expansion to high-latitude regions (*Jones et al., 2008*; *Jones et al., 2016*). While these mid-latitude cereals have been extensively studied, investigations of crops originated from other climate zones are needed. Using the South Asian (SA) legume mungbean as a test case, here, we investigate how climatic adaptation might affect crop spread route and the evolutionary changes making such spread possible.

Mungbean (*Vigna radiata* [L.] Wilczek var. *radiata*), also known as green gram, is an important grain legume in Asia (*Nair and Schreinemachers, 2020*), providing carbohydrates, protein, folate, and iron for local diets and thereby contributing to food security (*Kim et al., 2015*). Among pulses, mungbean is capable of tolerating moderate drought or heat stress and has a significant role in rainfed agriculture across arid and semiarid areas (*Pratap et al., 2019*), which are likely to have increased vulnerabilities to climate change. Although there have been studies about the genetic diversity of cultivated and wild mungbean (*Ha et al., 2021*; *Kang et al., 2014*; *Noble et al., 2018*; *Sangiri et al., 2007*), the evolutionary history of cultivated mungbean after domestication still lacks genetic studies. Existing archeological evidence suggests that South Asia is the probable area of mungbean domestication, and at least two independent domestication events have been suggested, including Maharashtra and the eastern Harappan zone (*Fuller and Harvey, 2006*). The early archeological records suggest that the selection of large seed sizes occurred in the eastern Harappan zone by the third millennium BC and in Maharashtra, dating to the late second to early first millennium BC (*Fuller and Harvey, 2006*). This pulse later spread to mainland Southeast Asia and has been reported in southern Thailand dating

to the late first millennium BC (*Castillo et al., 2016*). Further north, the earliest record of mungbean in China was from the book Qimin Yaoshu (齊民要術, 544 AD). While mungbean is also cultivated in Central Asia today, it was not identified in archaeobotanical evidence ranging from several millennium BC to the medieval period (*Miller, 1999*; *Spengler et al., 2018b*; *Spengler et al., 2017*), suggesting later arrival. While the archaeobotanical studies elucidated the route of mungbean cultivation range expansion, researches are still needed to identify the genetic evidence and factors shaping such spread route.

A recent genetic study revealed that present-day cultivated mungbeans have the same haplotype in the promoter region, reducing the expression of *VrMYB26a* (*Lin et al., 2022*), a candidate gene controlling the important domestication trait, pod shattering, in several *Vigna* species (*Takahashi et al., 2020*). This suggests the loss of pod-shattering phenotype in cultivated mungbean may have a common origin and despite the archaeobotanical findings of several independent early cultivations of mungbean in South Asia (*Fuller and Harvey, 2006*), descendants from one of these cultivation origins might have dominated South Asia before the pan-Asia expansion. Since large regions remain archaeologically unexplored, utilization of genetic data can be a crucial complementation to reconstruct crop evolutionary history. Using seed proteins (*Tomooka et al., 1992*) and isozymes (*Dela Vina and Tomooka, 1994*), previous studies proposed two expansion routes out of India, one in the south to Southeast Asia and the other in the north along the silk road to China. While later studies used DNA markers to investigate mungbean population structure (*Breria et al., 2020*; *Gwag et al., 2010*; *Islam and Blair, 2018*; *Noble et al., 2018*; *Sandhu and Singh, 2021*; *Sangiri et al., 2007*), few have examined these hypothesized routes in detail. Therefore, genomic examination of the cultivation rage expansion proposed by archaeobotanical studies and the elucidation of its contributing factors are strongly needed.

In this study, we compiled an international effort, reporting a global mungbean diversity panel of more than 1100 accessions derived from (i) the mungbean mini-core collection of the World Vegetable Center (WorldVeg) genebank, (ii) the Australian Diversity Panel (ADP), and (iii) the Vavilov Institute (VIR), which hosts a one-century-old collection enriched with mid-latitude Asian accessions that are underrepresented in other genebanks, many of which were old landraces collected by Nikolai I. Vavilov and his teams in the early 20th century (*Burlyaeva et al., 2019*). These germplasms harbor a wide range of morphological variations (*Figure 1A*) and constitute the most comprehensive representation of worldwide mungbean genetic variation. We used this resource to investigate the global history of mungbean after domestication to reveal a spread route highly affected by climatic constraints across Asia, eventually shaping the phenotypic characteristics for local adaptation to distinct environments.

## Results

### Population structure and spread of mungbean

Using DArTseq, we successfully obtained new genotype data of 290 mungbean accessions from VIR *Supplementary file 1a*. Together with previous data (*Breria et al., 2020*; *Noble et al., 2018*), our final set included 1108 samples with 16 wild and 1092 cultivated mungbean. A total of 40,897 SNPs were obtained. Of these, 34,469 bi-allelic SNPs, with a missing rate less than 10%, were mapped on 11 chromosomes and retained for subsequent analyses.

The genetic structure was investigated based on the 10,359 LD-pruned SNPs. Principal component analysis (PCA, *Figure 1C*) showed a triangular pattern of genetic variation among cultivated mungbeans, consistent with previous studies (*Breria et al., 2020*; *Noble et al., 2018*; *Sokolkova et al., 2020*) and ADMIXTURE K=3 (*Figure 1B*). The geographic distribution of these genetic groups is not random, as these three groups are distributed in South Asia (India and Pakistan), Southeast Asia (Cambodia, Indonesia, Philippines, Thailand, Vietnam, and Taiwan), and more northernly parts of Asia (China, Korea, Japan, Russia, and Central Asia). As K increased, the cross-validation (CV) error decreased a little after K=4 (*Figure 1—figure supplement 1*), where the north group could be further divided (*Figure 1B*). Therefore, worldwide diversity of cultivated mungbean could be separated into four major genetic groups corresponding to their geography: SA, Southeast Asian (SEA), East Asian (EA), and Central Asian (CA) groups. Note that the genetic groups were named after the region where most of their members distribute, and exceptions exist. For example, many EA accessions also distribute in Central Asia, and some SEA accessions were found near the eastern and northeastern

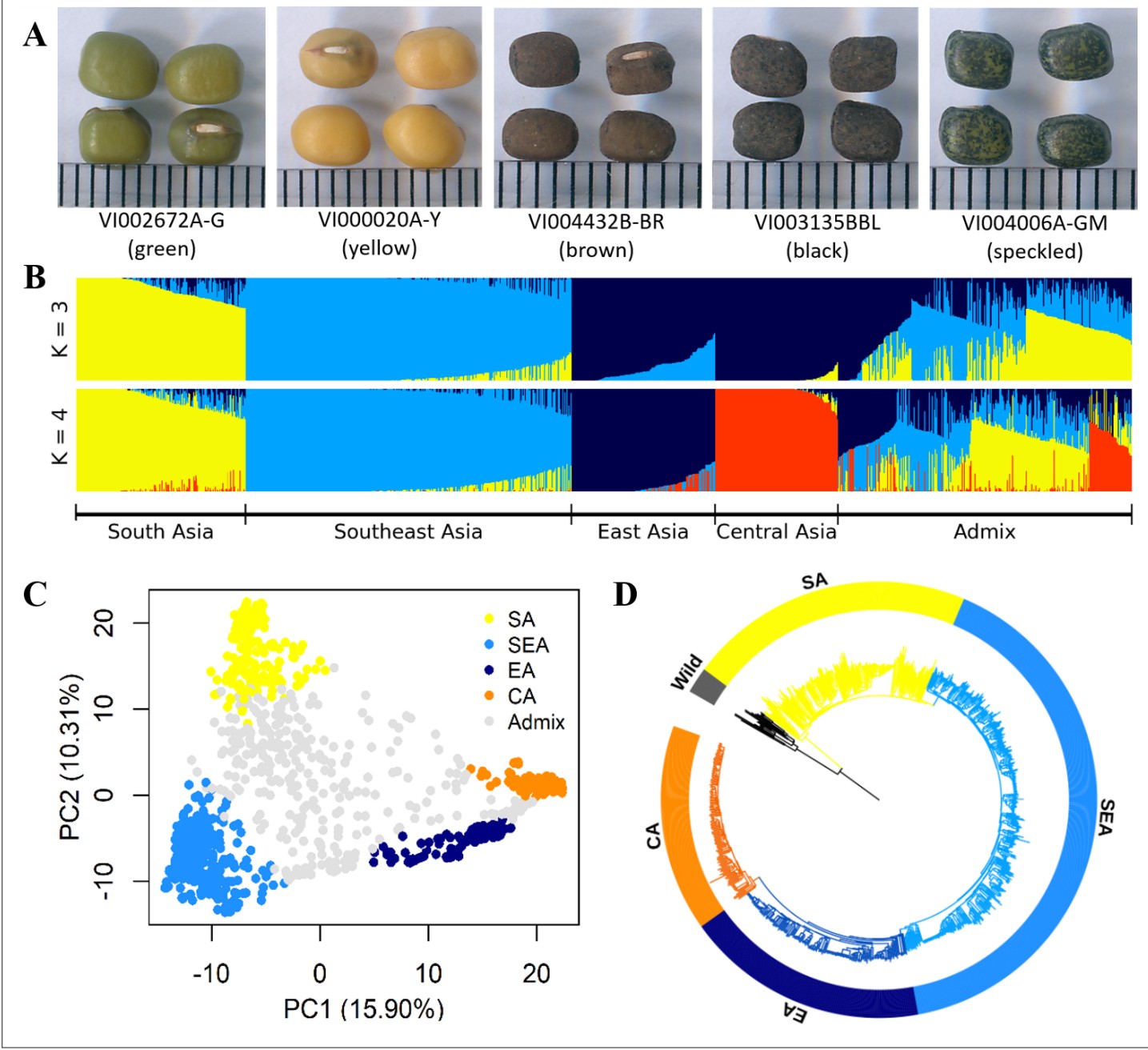

**Figure 1.** Diversity of worldwide mungbean. (**A**) Variation in seed color. (**B**) ADMIXTURE ancestry coefficients, where accessions were grouped by group assignments (Q≥0.7). (**C**) Principal component analysis (PCA) plot of 1092 cultivated mungbean accessions. Accessions were colored based on their assignment to four inferred genetic groups (Q≥0.7), while accessions with Q<0.7 were colored gray. (**D**) Neighbor-joining (NJ) phylogenetic tree of 788 accessions with Q≥0.7 with wild mungbean as outgroup (black color).

The online version of this article includes the following figure supplement(s) for figure 1:

**Figure supplement 1.** Cross-validation (CV) errors of ADMIXTURE.

coasts of India. Throughout this work, we make clear distinction between genetic group names (e.g. SA) and a geographic region (e.g. South Asia). Therefore, unlike any other previous work in this species, this study incorporates global genetic variation among cultivated mungbean of this important crop.

Using wild progenitor *V. radiata* var. *sublobata* (Wild hereafter) as the outgroup, the accession- (*Figure 1D*) and population-level (*Figure 2A*) phylogenies both suggest CA to be genetically closest to EA. The SEA group is more distant, and SA is the most diverged. This relationship is supported

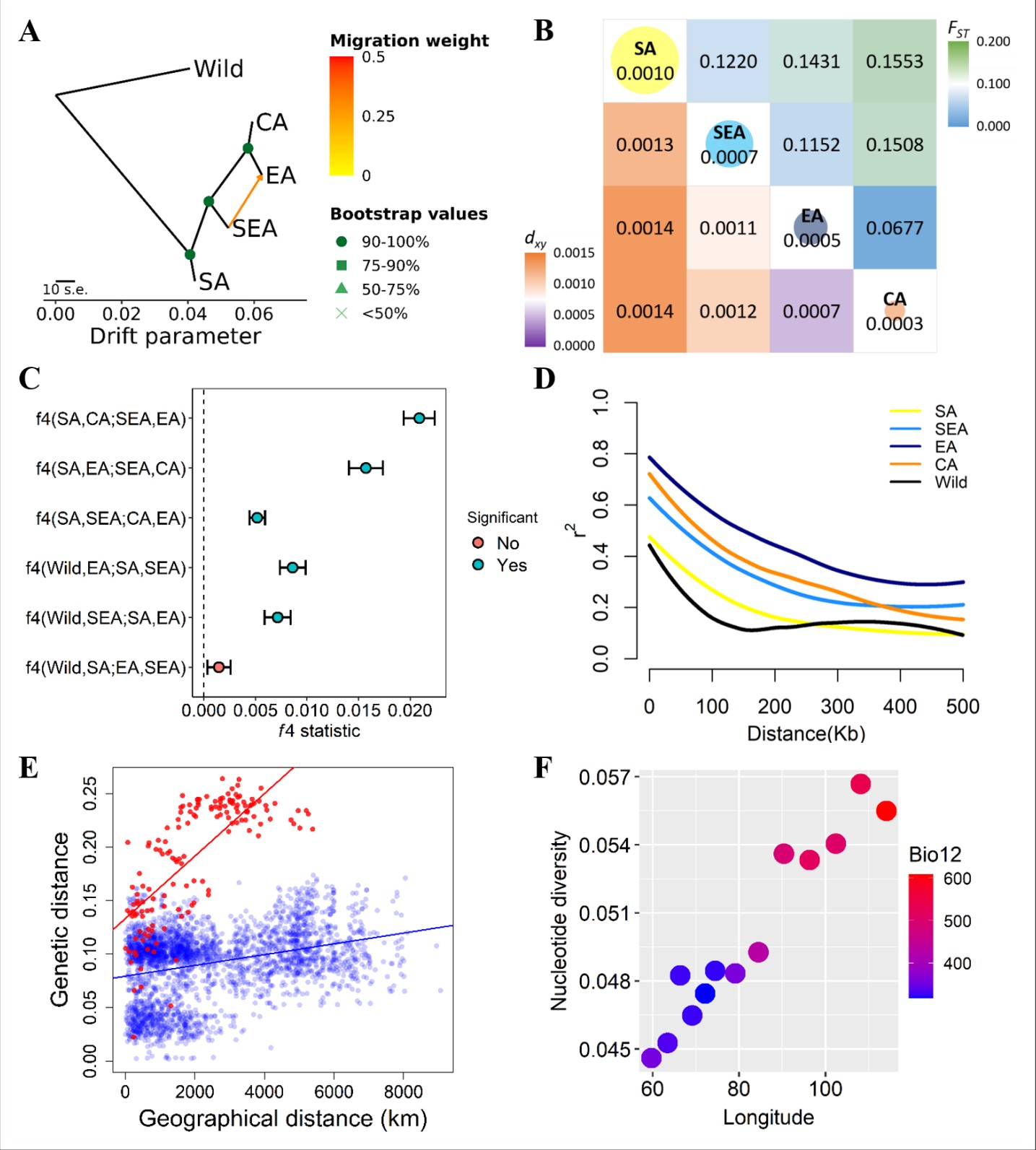

**Figure 2.** Fine-scale genetic relationship and admixture among four inferred genetic groups. (**A**) TreeMix topologies with one suggested migration event. Colors on nodes represent support values after 500 bootstraps. (**B**) Diversity patterns within and between inferred genetic groups as estimated using nucleotide diversity ($\pi$ in diagonal, where the size of the circle represents the level of $\pi$) and population differentiation ($F_{ST}$ in upper diagonal and $d_{xy}$ in lower diagonal). (**C**) $f4$ statistics. Points represent the mean $f4$ statistic, and lines are the SE. Only $f4$ statistics with Z-score>|3| are considered

*Figure 2 continued on next page*

*Figure 2 continued*

statistically significant. The dashed line denotes *f*4=0. (**D**) Linkage disequilibrium (LD) decay. (**E**) Isolation by distance plot of genetic distance versus geographic distance, with the southern group in red circles and the northern group in blue circles. (**F**) Relationship between Bio12 (annual precipitation) and nucleotide diversity (π) of the East Asian (EA) genetic group across the east-west axis of Asia. Dot colors represent the annual precipitation of each population.

The online version of this article includes the following figure supplement(s) for figure 2:

**Figure supplement 1.** Schematic representation to investigate presence of admixture in a target population from two source populations using admixture *f*3 statistic.

**Figure supplement 2.** Estimates of divergence time and inferred mungbean movement over time across Asia.

by the outgroup *f*3 tests showing CA shared the highest level of genetic drift with EA, followed by SEA and SA (***Supplementary file 1b***). Pairwise $F_{ST}$ and $d_{xy}$ also give the same conclusion (***Figure 2B***). Similarly, the *f*4 tests (***Figure 2C***) strongly reject the cases where SEA and CA form a clade relative to SA and EA (*f*4[SA,EA;SEA,CA]=0.016, *Z*=9.519) or SEA and EA form a clade relative to SA and CA (*f*4[SA,CA;SEA,EA]=0.021, *Z*=13.956), again suggesting EA and CA to be closest. With regards to the relationship among Wild, SA, SEA, and EA, *f*4 tests suggest SEA and EA form a clade relative to Wild and SA (non-significant results in *f*4[Wild,SA;EA,SEA] but opposite in other combinations). Notably, both TreeMix (***Figure 2A***) and the *f*4 test (***Figure 2C***, *f*4[SA,SEA;CA,EA]=0.005, *Z*=6.843) suggest gene flow between SEA and EA. Consistent with archeological evidence of SA domestication, the nucleotide diversity (π) decreased from SA ($1.0\times10^{-3}$) to SEA ($7.0\times10^{-4}$) and EA ($5.0\times10^{-4}$), while the CA group has lowest diversity ($3.0\times10^{-4}$; ***Figure 2B***). Linkage disequilibrium (LD) also decays the fastest in Wild and then the SA group (***Figure 2D***), followed by other genetic groups. In summary, all analyses are consistent with our proposed order of cultivated mungbean divergence.

Our proposed demographic history could be confounded by factors such as complex hybridization among groups. For example, SEA and CA might have independently originated from SA and later generated a hybrid population in EA (***Figure 2—figure supplement 1A***). Other possibilities are that either SEA or CA is the hybrid of other populations (***Figure 2—figure supplement 1B and C***). We examined these possibilities using *f*3 statistics for all possible trios among the four groups. None of the tests gave a significantly negative *f*3 value (***Supplementary file 1c***), suggesting the lack of a strong alternative model to our proposed relationship among these four groups.

Based on the solid relationship among these genetic groups, we used fastsimcoal2 to model their divergence time, allowing population size change and gene flow at all time points (***Figure 2—figure supplement 2A–D***). According to this model, after initial domestication, the out-of-India event (when other groups diverged from SA) happened about 8.3 thousand generations ago (kga) with 75% parametric bootstrap range between 4.7 and 11.3 kga. Not until more than 5000 generations later (2.7 kga, 75% range 1.1–4.6 kga) did SEA diverge from the common ancestor of present-day EA and CA. CA diverged from EA only very recently (0.2 kga, 75% range 0.1–0.8 kga). Note that the divergence time was estimated in the number of generations, and the much longer growing seasons in the southern parts of Asia may allow more than one cropping season per year (***Mishra et al., 2022***; ***Vir et al., 2016***).

Our results suggest the non-SA accessions have a common origin out of India (otherwise these groups would branch off independently from the SA group). Given this, the phylogenetic relationship (***Figure 2A***) is consistent with the following hypotheses. (1) The east hypothesis: mungbean expanded eastward and gave rise to the SEA group. This group might initially occupy northeast South Asia and later expanded to Southeast Asia either through the land or maritime route (***Castillo et al., 2016***; ***Fuller et al., 2011***). The group later expanded northward as EA. EA expanded westward into Central Asia and gave rise to the CA group. (2) The north hypothesis: the group leaving South Asia first entered Central Asia as the EA group. EA expanded eastward into East Asia through the Inner Asian Mountain Corridor (***Stevens et al., 2016***). The eastern population of EA expanded southward as the SEA group, and later the western population of EA diverged as the CA group. (3) The northeast hypothesis: the group leaving South Asia (through either of the above-mentioned routes) was first successfully cultivated in northern East Asia without previously being established in Southeast Asia or Central Asia. The EA group then diverged southward as SEA and later expanded westward, giving rise to CA. Consistent with this model, the genetic variation of the EA group gradually declines from east

to west, accompanied by the gentlest decline of precipitation per unit geographic distance across Asia (*Figure 2F*).

While all three hypotheses are consistent with the phylogeny (*Figure 2A*), the SEA group originated earlier than EA in the east hypothesis but later in the two other hypotheses. The former case predicts higher nucleotide diversity and faster LD decay in SEA than EA, which is supported by our results (*Figure 2B and D*). While populations that were established in a region for an extended time could accumulate genetic differentiation, generating patterns of isolation by distance, rapid-spreading populations in newly colonized regions could not (*Lee et al., 2017*; The 1001 *1001 Genomes Consortium, 2016*). Using this idea, Mantel's test revealed a significantly positive correlation between genetic and geographic distances for the SA genetic group (*r*=0.466, *P*=0.010), followed by SEA (*r*=0.252, although not as significant, *P*=0.069). No such association was found for EA (*r*=0.030, *P*=0.142) or CA (*r*=0.087, *P*=0.172). In addition, the southern groups (SA and SEA) together (*r*=0.737, *P*=0.001) have a much stronger pattern of isolation by distance than the northern groups (EA and CA, *r*=0.311, *P*=0.001; *Figure 2E*). Using Q≥0.5 instead of Q≥0.7 to assign individuals into genetic groups generated results that are largely consistent (*Supplementary file 1d*). These results are again consistent with the 'east hypothesis' that local accessions from the SA and SEA groups were established much earlier than those from EA and CA. Finally, the genetic variation of the EA group is highest in the eastern end and declines westward (*Figure 2F*). This does not support the north hypothesis where EA first existed in Central Asia and expanded eastward.

## Environmental differentiation of the inferred genetic groups

We further examined the possible causes governing the expansion of mungbean cultivation ranges. For a crop to be successfully cultivated in a new environment, dispersal and adaptation are both needed. Being a crop that has lost the ability of pod shattering, the spread of mungbean was governed by commerce or seed exchange. While barriers such as the Himalayas or Hindu Kush may limit human activity, South and Central Asia was already connected by a complex exchange network linking the north of Hindu Kush, Iran, and the Indus Valley as early as about 4 thousand years ago (kya; *Dupuy, 2016*; *Kohl, 2007*; *Kohl and Lyonnet, 2008*; *Lamberg-Karlovsky, 2002*; *Lombard, 2020*; *Lyonnet, 2005*), and some sites contain diverse crops originated across Asia (*Spengler et al., 2021*). Similarly, other ancient land or maritime exchange routes existed among South, Southeast, East, and Central Asia (*Stevens et al., 2016*). This suggests that mungbean could have been transported from South to Central Asia, but our genetic evidence suggests that the present-day CA group did not descend directly from the SA group. Therefore, we investigated whether climatic adaptation, that is, the inability of mungbean to establish in a geographic region after human-mediated long-range expansion, could be a contributing factor.

Multivariate ANOVA (MANOVA) of eight bioclimatic variables (after removing highly-correlated ones; *Supplementary file 1e,f*) indicated strong differentiation in the environmental niche space of the four genetic groups (*Supplementary file 1g,h*). PCA of climatic factors clearly reflects geographic structure, where the axis explaining most variation (PC1, 42%) separates north and south groups and is associated with both temperature- and precipitation-related factors (*Figure 3A* and *Supplementary file 1i*). Consistent with their geographic distribution, overlaps between EA and CA and between SA and SEA were observed. While these analyses were performed using bioclimatic variables from year-round data, we recognized that summer is the cropping season in the north. Parallel analyses using the temperature and precipitation of May, July, and September yielded similar results (*Supplementary file 1j*; *Figure 3—figure supplement 1*).

Based on the Köppen climate classification (*Köppen, 2011*), we categorized the Asian mungbean cultivation range into six major climate zones (*Figure 3—figure supplement 2*): dry hot (BSh and BWh), dry cold (BSk and BWk), temperate dry summer (Csa), tropical savanna (Aw), continental (Dwb and Dfb), and temperate wet summer (Cfa and Cwa). The former three are relatively drier than the latter three zones. While SEA and CA are relatively homogeneous, SA and EA have about half of the samples in the dry and non-dry zones (*Figure 3—figure supplement 2*). We, therefore, separated SA into SAe and SAw and EA into EAe and EAw, corresponding to the wetter eastern and drier western regions within the SA and EA ranges. Environmental niche modeling revealed distinct suitable regions of these six groups except for CA and EAw, whose geographical ranges largely overlap (*Figure 3B*). Consistent with PCA, pairwise Schoener's D values are smallest between the northern and southern

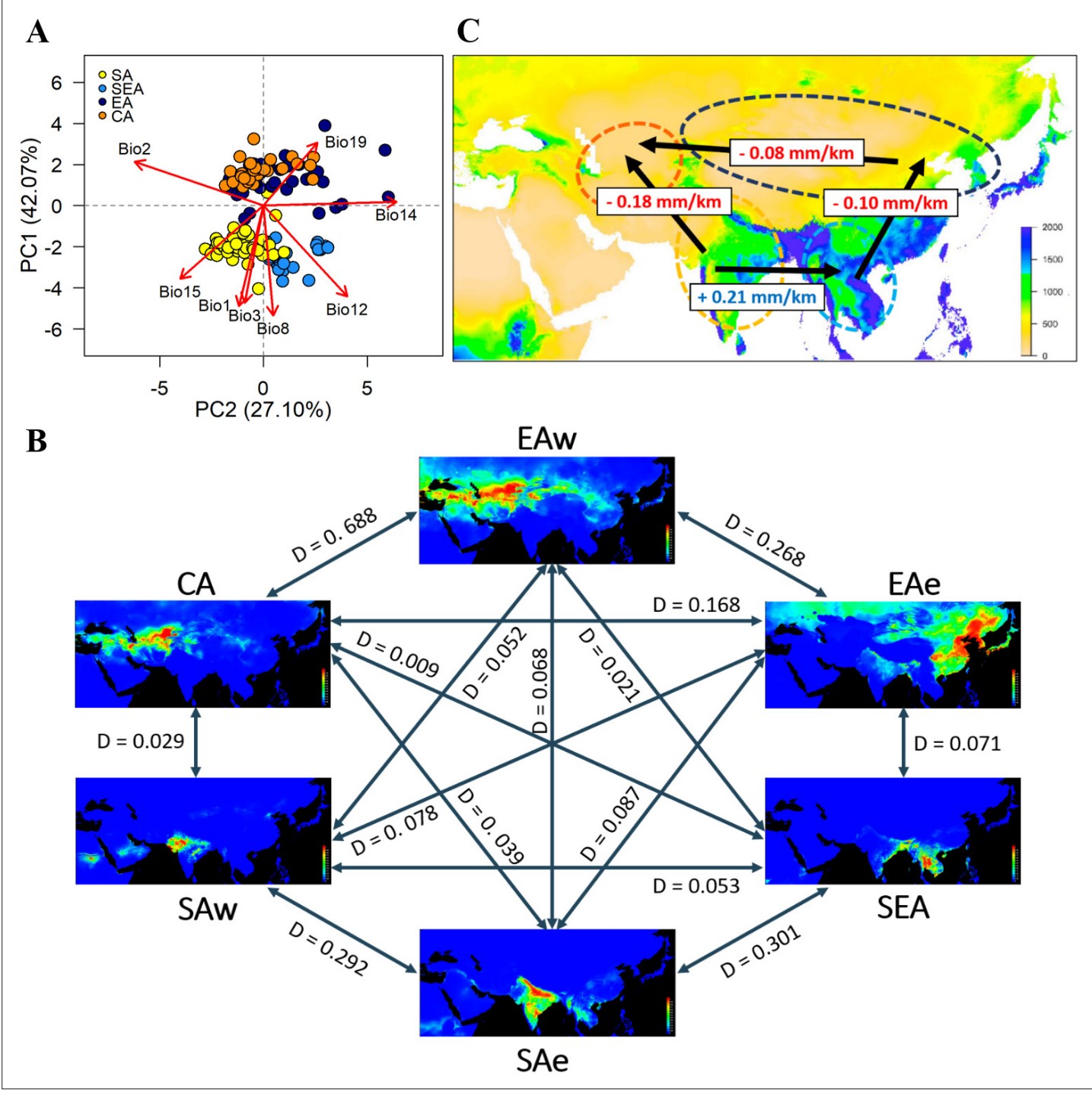

**Figure 3.** Environmental variation among genetic groups of mungbean. (**A**) Principal component analysis (PCA) of the eight bioclimatic variables. Samples are colored according to four inferred genetic groups as indicated in the legend. (**B**) Predicted distribution at current climate conditions. Red color indicates high suitability, and blue indicates low suitability. Values between pairs represent niche overlap measured using Schoener's D, and higher values represent higher overlaps. Abbreviations: SAw: South Asia (west), SAe: South Asia (east); SEA: Southeast Asia; EAe: East Asia (east); EAw: East Asia (west), and CA: Central Asia. (**C**) Environmental gradient across potential directions of expansion. The value on each arrow indicates a change in annual precipitation per kilometer. The background map is colored according to annual precipitation (Bio12, in mm).

The online version of this article includes the following figure supplement(s) for figure 3:

**Figure supplement 1.** Principal component analysis (PCA) of the growing season climatic data including temperature and precipitation of May, July, and September.

*Figure 3 continued on next page*

*Figure 3 continued*

**Figure supplement 2.** The distribution of accessions in major climate zones according to the Köppen climate classification (**Köppen, 2011**).

**Figure supplement 3.** Predicted distributions of six groups based on monthly temperature and precipitation (May, July, and September) during the summer growing season.

**Figure supplement 4.** Monthly temperature and precipitation variations among the four genetic groups.

**Figure supplement 5.** Environmental gradient across Asia.

groups while largest (suggesting overlaps of niche space) between the eastern and western subsets within north and south (**Figure 3B**), consistent with PCA that the major axis of climatic difference is between the northern and southern parts of Asia. Analyses using temperature and precipitation from May, July, and September yielded similar results (**Figure 3—figure supplement 3**). Given a single out-of-India event (**Figure 2A**), the results suggest it might be easier to first cultivate mungbean in Southeast rather than Central Asia, supporting the east hypothesis.

While both temperature and precipitation variables differ strongly between north and south, one should note that these year-round temperature variables do not correctly reflect conditions in the growing seasons. In the north, mungbean is mostly grown in summer where the temperature is close to the south (**Figure 3—figure supplement 4A–C**). On the other hand, precipitation differs drastically between the north and south, especially for the CA group, where the summer-growing season is the driest of the year (**Figure 3—figure supplement 4D**). By estimating the regression slope of annual precipitation on geographical distance, we obtained a gradient of precipitation change per unit geographic distance between pairs of genetic groups (**Figure 3C**). Despite the SA-SEA transect having the steepest gradient (slope = 0.21), the spread from SA to SEA has been accompanied by an increase of precipitation and did not impose drought stress. However, the second highest slope (0.18) is associated with a strong precipitation decrease if the SA group were to disperse to Central Asia. Results from the precipitation of May, July, and September yielded similar conclusion (**Figure 3—figure supplement 5**). This likely explains why no direct historic spread is observed from South to Central Asia.

## Trait variation among genetic groups

If environmental differences constrained the spread route of mungbean, the currently cultivated mungbean accessions occupying distinct environments should have locally adaptive traits for these environments. Indeed, PCA of four trait categories shows substantial differences among genetic groups (phenology, reproductive output, and size in field trials, as well as plant weight in lab hydroponic systems, **Figure 4A**). In the field, CA appears to have the shortest time to flowering, the lowest yield in terms of seed size and pod number, and the smallest leaf size (**Figure 4B** and **Supplementary file 1k**). On the other hand, SEA accessions maximize seed size, while SA accessions specialize in developing the largest number of pods (**Figure 4B**). These results suggest that CA has a shorter crop duration, smaller plant size, and less yield, consistent with drought escape phenotypes. This is consistent with the northern short-growing season constrained by temperature and daylength (below), as well as the low precipitation during the short season.

In terms of seedling response to drought stress, the $Q_{ST}$ values of most traits (root, shoot, and whole plant dry weights under control and drought treatments) are higher than the tails of SNP $F_{ST}$, suggesting trait evolution driven by divergent selection (**Figure 4C**; **Figure 4—figure supplement 1**). Significant treatment, genetic group, and treatment by group interaction effects were observed except on a few occasions (**Table 1**). Consistent with field observation, SEA has the largest seedling dry weight (**Figure 4D**). While simulated drought significantly reduced shoot dry weight (SDW) for all groups, the effect on SEA is especially pronounced (treatment-by-group interaction effect, $F_{2,575}$ = 23.55, $P<0.001$, **Table 1** and **Figure 4D**), consistent with its native habitats with abundant water supply (**Figure 3—figure supplement 4D** and **Supplementary file 1l**). All groups react to drought in the same way by increasing root:shoot ratio (**Figure 4D**), suggesting such plastic change may be a strategy to reduce transpiration. Despite the lack of treatment-by-group interaction ($F_{2,575}$ = 1.39, $P>0.05$), CA consistently exhibits a significantly higher root:shoot ratio, a phenotype that is potentially adaptive to its native environment of lower water supply (**Figure 3—figure supplement 4D** and **Supplementary file 1l**).

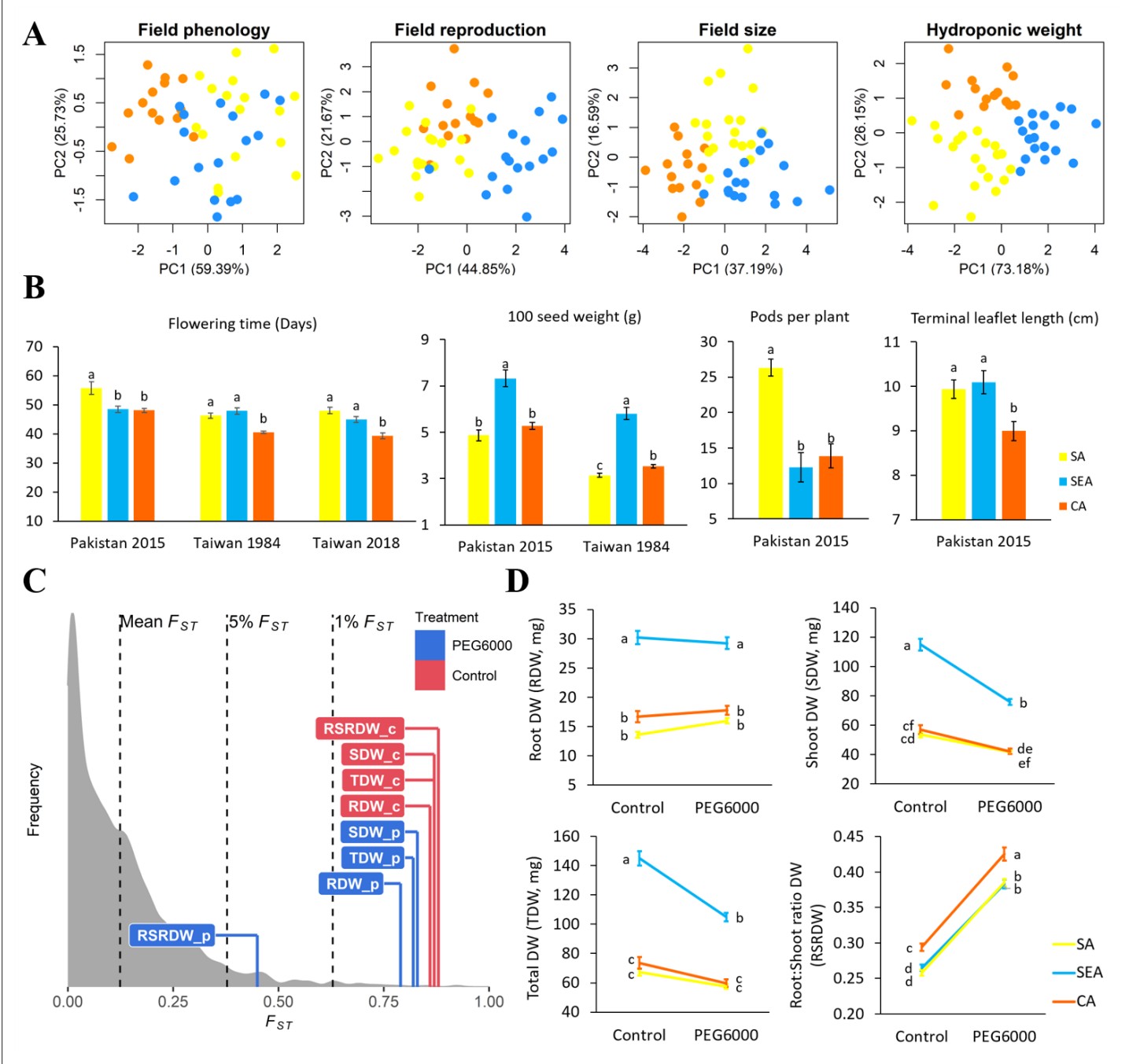

**Figure 4.** Quantitative trait differentiation among genetic groups. (**A**) Principal component analysis (PCA) of four trait categories. (**B**) Trait variability from common gardens in field experiments. Sample size of SA, SEA, and CA are 18, 17, and 14, respectively. (**C**) Comparison of $Q_{ST}$-$F_{ST}$ for four drought-related traits under two environments. $F_{ST}$ values (mean, 5%, and 1%) were indicated by black dashed lines. The $Q_{ST}$ for each trait was colored according to treatment and was calculated as Equation 2 in Materials and methods. Abbreviations: RDW: root dry weight; SDW: shoot dry weight; TDW: total dry weight; RSRDW: root:shoot ratio dry weight; c: control; p: PEG6000. (**D**) Effect of PEG6000 (–0.6 MPa) on RDW, SDW, TDW, and RSRDW among genetic groups. Sampe size of SA, SEA, and CA are 20, 18, and 14, repectively. Data were expressed as the mean ± SE. Lowercase letters denote significant differences under Tukey's honestly significant difference test in (**B**) and (**D**).

The online version of this article includes the following figure supplement(s) for figure 4:

**Figure supplement 1.** Comparison of $Q_{ST}$-$F_{ST}$ for four drought-related traits under two environments.

**Table 1.** ANOVA *F* values for the dry weight (mg) of mungbean seedlings across three different genetic groups.

| Source of variation | Degrees of freedom (df) | Root dry weight | Shoot dry weight | Total dry weight | Root:shoot ratio dry weight |
|---|---|---|---|---|---|
| Treatment | 1 | 2.65[n.s.] | 133.26*** | 72.26*** | 978.76*** |
| Genetic group | 2 | 60.63*** | 79.62*** | 76.54*** | 13.27*** |
| Treatment × Genetic group | 2 | 3.29* | 23.55*** | 17.79*** | 1.39[n.s.] |

*$P<0.05$ and ***$P<0.001$; n.s. non-significant.

### Support from ancient Chinese sources

Mungbean has been occasionally mentioned in ancient Chinese sources. Here, we report the records associated with our proposed mungbean spread route and the underlying mechanisms. The 'Classic of Poetry' (Shijing 詩經) contains poems dating between the 11th and 7th centuries BCE near the lower and middle reaches of the Yellow River. While crops (especially soy bean, 菽), vegetables, and many other plants have been mentioned, mungbean was not recorded. This is consistent with our results that mungbean had not reached the northern parts of East Asia at that time (the EA group diverged from the SEA group at around 2.7 kga). The first written record of mungbean in China is in an agricultural encyclopedia Qimin Yaoshu (齊民要術, 544 AD, Chinese text and translation in Supplementary note), whose spatiotemporal background (~1.5 kya near the lower reaches of Yellow River) is again consistent with our estimated origin of the EA group.

Our results suggest that the expansion of the mungbean cultivation range may be associated with the novel phenotypic characteristics potentially adaptive to the new environments. This proposal would be rejected if the novel phenotypic characteristics appeared very recently. In support of our proposal, Xiangshan Yelu (湘山野錄, an essay collection during 1068–1077 AD) recorded that mungbean from the southern parts of Asia had higher yield and larger grains than those in northern China (Chinese text and translation in Supplementary note). Similarly, Tiangong Kaiwu (天工開物, 1637 AD) mentioned that mungbean must be sown during July and August (Chinese text and translation in Supplementary note). The record suggests that the daylength requirement restricts the sowing period of mungbean in the north. Together with the dry summer (*Figure 3—figure supplement 4D*) and soon-arriving autumn frost, there might be a strong selection favoring accessions with the rapid life cycle. These records suggest the phenotypic characteristics of northern accessions did not originate very recently, and the unique distribution of climatic zones in Asia resulted in not only the specific patterns of expansion but also the evolution of novel phenotypic characteristics in mungbean.

## Discussion

Using mungbean as a test case, we combined population genomics, environmental niche modeling, empirical field and laboratory investigation, and ancient Chinese text analyses to demonstrate the importance of climatic adaptation in dictating the unique patterns of cultivation range expansion after domestication. In this study, we focus on how or when mungbean could be established as part of local agriculture throughout Asia. We showed that after leaving South Asia, mungbean was likely first cultivated in Southeast Asia, East Asia, and finally Central Asia. We acknowledge that our data do not allow us to specify the number of previous out-of-India events that did not leave traces in modern genetic data or their exact routes (e.g. whether mungbean expanded from South to Southeast Asia through the land or maritime routes). While there might be multiple attempts to bring mungbean out of India as a commodity for consumption, our results suggest all present-day non-SA accessions have a common out-of-India origin.

### The climate-driven spread route despite historical human activities

Combining archeological records, population genetics, and niche modeling (*Figures 2 and 3*), our results suggest that after the early cultivation of mungbean in northwestern or southern South Asia (*Fuller, 2007*; *Kingwell-Banham et al., 2015*), the large environmental difference may restrict its northward spread to Central Asia. Mungbean may first spread to eastern South Asia, and the subsequent expansion to Southeast Asia might be facilitated by the environmental similarity between these

two regions. This is supported by archaeobotanical remains from the Thai-Malay Peninsula date to ca. 400–100 BCE (*Castillo et al., 2016*). It took more than 5000 generations until mungbean further spread to northeast Asia, again likely due to the environmental difference. The later appearance of mungbean in northern China is also supported by historical records. After that, the EA group spread across the northern part of Asia within a few thousand generations. Our proposed route suggests that mungbean reached Central Asia at the latest, consistent with its absence from archeological sites in Central Asia, including Turkmenistan and Uzbekistan in the Chalcolithic and Bronze ages (fifth to second millennium BC; *Miller, 1999*), Southeastern Kazakhstan in the Iron age dating first millennium BC (*Spengler et al., 2017*), and eastern Uzbekistan during the medieval period (800–1100 AD; *Spengler et al., 2018b*). In addition, mungbean was only mentioned later by the 18th and early 19th centuries as a pulse grown in the Khiva region of Uzbekistan (*Annanepesov and Bababekov, 2003*).

In this study, we suggest that the ability to disperse may not be an essential factor restricting mungbean spread from South to Central Asia. Cultivated mungbean has lost the natural ability of pod shattering to disperse seeds, and they mostly traveled through landscapes by human-mediated seed exchange or commerce. Evidence of long-distance human-mediated dispersal of mungbean was available. For example, mungbean seeds have been found near the Red Sea coast of Egypt during the Roman (AD 1–250) period (*Van der Veen and Morales, 2015*). As early as about 4 kya, the Bactria–Margiana Archaeological Complex civilization north of the Hindu Kush had extensive contact with the Indus Valley Civilization (*Dupuy, 2016*; *Kohl, 2007*; *Kohl and Lyonnet, 2008*; *Lamberg-Karlovsky, 2002*; *Lombard, 2020*; *Lyonnet, 2005*). By 1500 BC, the 'Trans-Eurasian Exchanges' of major cereal crops has happened (*Liu et al., 2019*). The frequent crop exchange is evidenced by archaeobotanical findings in the Barikot site (ca. 1200 BC-50 AD) in northern Pakistan (*Spengler et al., 2021*), where diverse crops were cultivated, including those from West Asia (wheat, barley, pea, and lentil), South Asia (urdbean/mungbean), and likely East Asia (rice). Despite this, in Bronze-age archeological sites north of Hindu Kush, legumes (such as peas and lentils) were observed to a lesser extent than cereals, and SA crops were not commonly found (*Jeong et al., 2019*; *Spengler, 2015*; *Spengler et al., 2014a*; *Spengler et al., 2018a*; *Spengler et al., 2014b*). Interestingly, archeologists suggested legume's higher water requirement than cereals may be associated with this pattern, and pea and lentil's role as winter crops in Southwest Asia may be associated with their earlier appearance in northern Central Asia than other legumes (*Spengler et al., 2014a*; *Spengler et al., 2018a*; *Spengler et al., 2014b*). Therefore, despite the possibility of human-mediated seed dispersal between South and Central Asia, our results and archeological evidence concurred that mungbean arrived in Central Asia at the latest, likely restricted by environmental adaptation.

## Local adaptation of mungbean genetic groups

Despite the profound impact of human-mediated dispersal on the spread of these and many other crops (*Herniter et al., 2020*; *Kistler et al., 2018*), in mungbean, we suggest adaptation to distinct climatic regimes to be an important factor in the establishment after dispersal. Mungbean is commonly grown under rainfed cultivation and depends on the residual moisture in the fields after the primary crop, thus responding to water stress (*Douglas et al., 2020*). In the south, a temperature range of 20–30°C and annual precipitation of 600–1000 mm is optimal for mungbean (*Ha and Lee, 2019*). In Central Asia, however, the annual precipitation could be as low as 286 mm, greatly below the lower limit required for the southern mungbean. This situation could be further acerbated by the fact that mungbean might not be a highly valued crop under extensive care during cultivation. Indeed, the earliest record of mungbean in China (Qimin Yaoshu 齊民要術, 544 AD) emphasizes its use as green manure. In Central Asia, mungbean is a minor crop (*Rani et al., 2018*) grown with little input, only in the short duration between successive planting of main crops (which is also the dry season in Central Asia, *Supplementary file 1* and *Figure 3—figure supplement 4*) and using residual soil moisture with little irrigation. We suggest that the lack of extensive input subjects mungbean to more substantial local climatic challenges than highly valued high-input crops that receive intensive management, including irrigation. Therefore, the combination of climatic constraints and cultural usage, instead of physical barriers, may have shaped the historical spread route of the mungbean despite extensive human activities across the continent.

In addition to the constraint of soil moisture, other factors may have contributed to the selection of short-season accessions in the north. In the short summer seasons of much of Central Asia, short

crop cycling is a requirement. In Uzbekistan, mungbean is often sown in early July after the winter wheat season and harvested before mid-October to avoid delays in the next round of winter wheat and escape frost damage. Therefore, fast-maturing accessions are essential for this production system (*Rani et al., 2018*). Similar rotation systems using mungbean to restore soil fertility during the short summer season after the harvest of the main crop were also mentioned in ancient Chinese sources (*Chen, 1980*). Mungbean is a short-day species from the south, and daylength likely limits the window when mungbean could be grown in the north: Chinese texts during the 17th century (Tiangong Kaiwu 天工開物, 1637 AD) specifically mentioned the suitable duration to sow mungbean to control the flowering behavior for maximum yield (Supplementary note). Therefore, unlike in the south where yield appears to be an important selection target, the unique combination of daylength, agricultural practices, soil water availability, and frost damage in the north requires the selection for short-season accessions, likely limiting the direct adoption of southern accessions in the north. Consistent with this, CA accessions have a faster life cycle potentially adaptive to both short growing season and reduced soil water availability, with reduced plant size and lower yield as tradeoffs. These accessions also have increased root:shoot ratio for drought adaptation, similar to findings in rice (*Xu et al., 2015*), alfalfa (*Zhang et al., 2018*), and chickpea (*Kumar et al., 2012*).

About accession sampling and climatic niche modeling, we recognize that not all samples have available spatial data, and we do not have samples from some parts of Asia. For example, while most samples of the SEA group were collected from Taiwan, Thailand, and Philippines, we do not have many samples from the supposed contact zone between SA and SEA (Bangladesh and Myanmar) or between SEA and EA (southern China). If more samples were available from these contact zones, the modeled niche space between SA and SEA and between SEA and EA would be even more similar than the current estimate, strengthening our hypothesis that niche similarity might facilitate the cultivation expansion. On the other hand, clear niche differentiation between SA and CA was evident despite the dense sampling near their contact zone. Based on the Köppen climate classification, South Asia could be roughly separated into two major zones, with the eastern zone slightly more similar to Southeast Asia (*Figure 3—figure supplement 2*). This partially explained the existence of some SEA accessions in the northeastern coast of India. While the SEA genetic group was named after the geographic region where most of its members were found in the present time, we recognize the possibility that it first occupied northeastern South Asia when it diverged from SA. In that case, the SA-SEA divergence time (4.7–11.3 kga) might indicate the divergence between the two climate zones within South Asia rather than the expansion of mungbean into Southeast Asia, which may occur much later.

## Conclusion

Our study demonstrates that mungbean's cultivation range expansion is associated with climatic conditions, which shaped the genetic diversity and contributed to adaptive differentiation among genetic groups. The climatic differences likely also resulted in farmers' differential emphasis on using it mainly as a grain or green manure crop, further intensifying the phenotypic diversification among regional mungbean accessions that could be used as an invaluable genetic resource for genetic improvement in the future.

## Materials and methods
### Plant materials and SNP genotyping

A total of 290 cultivated mungbean (*V. radiata* var. *radiata*) accessions were provided by the VIR. Most of the accessions are mainly landraces collected during 1910–1960 and are considered these accessions as the oldest cultivated mungbean collection from VIR (*Burlyaeva et al., 2019*). The term landrace, as we use it here, refers to locally adaptive accessions coming from the countries traditionally cultivating them, which also lacks modern genetic improvement. The complete list of materials can be found in *Supplementary file 1a*. Genomic DNA was extracted from a single plant per accession using the QIAGEN Plant Mini DNA kit according to the manufacturer's instruction with minor modification of pre-warming the AP1 buffer to 65°C and increasing the incubation time of the P3 buffer up to 2 hr on ice to increase DNA yield. DNA samples were sent to Diversity Arrays Technology Pty Ltd, Canberra, Australia for diversity array technology sequence (DArTseq) genotyping.

DArTseq data of 521 accessions from the ADP (*Noble et al., 2018*) and 297 accessions from the WorldVeg mini-core (*Breria et al., 2020*) were also included in this study. In total, our dataset contains more than 1000 accessions (1092) and covers worldwide diversity of cultivated mungbean representing a wide range of variation in seed color (*Figure 1A*). Sixteen wild mungbean (*V. radiata* var. *sublobata*) accessions were included as an outgroup. While all accessions used in this study have the country of origin information, only those from VIR have detailed longitude and latitude information. Therefore, for analyses connecting genetic information and detailed location (the isolation by distance analyses), only the VIR samples were used.

The major goal of this study is to investigate the patterns of population expansion and the underlying ecological causes instead of detailed haplotype analyses of specific genomic regions. For this goal, genomewide SNPs provide similar information as whole-genome sequencing, as have been shown in other species. Compared to other genotyping-by-sequencing technologies, DArTseq has the additional advantage of less missing data among loci or individuals, providing a more robust estimation of population structure.

## SNP calling

Trimmomatic version 0.38 (*Bolger et al., 2014*) was used to remove adapters based on the manufacturer's adapter sequences. Reads for each accession were trimmed for low-quality bases with quality scores of Q≤10 using SolexaQA version 3.1.7.1 (*Cox et al., 2010*) and mapped to the mungbean reference genome (Vradiata_ver6, *Kang et al., 2014*) using the Burrows-Wheeler Aligner version 0.7.15 (*Li and Durbin, 2009*). Reads were then sorted and indexed using samtools version 1.4.1 (*Li et al., 2009*). We used Genome Analysis Toolkit (GATK) version 3.7–0-gcfedb67 (*McKenna et al., 2010*) to call all sites, including variant and invariant sites. We obtained 1,247,721 sites with a missing rate of <10% and a minimum quality score of 30. SNP calling was performed using GATK (*McKenna et al., 2010*). Finally, we used VCFtools version 0.1.13 (*Danecek et al., 2011*) to remove SNPs with more than two alleles and 10% missing data, resulting in 34,469 filtered SNPs. To reduce non-independence caused by LD among SNPs, SNPs were pruned based on a 50-SNP window with a step of five SNPs and $r^2$ threshold of 0.5 in PLINK (*Purcell et al., 2007*). This dataset of 10,359 LD-pruned SNPs (10% missing data) was applied for all analyses related to population genomics unless otherwise noted. For TreeMix that require LD-pruned SNPs with no missing dataset, we used 4396 LD-pruned SNPs with no missing data.

## Population genetics and differentiation analyses

Population structure was investigated based on 10,359 LD-pruned SNPs using ADMIXTURE (*Alexander et al., 2009*) with the number of clusters (K) ranging from 1 to 10. The analyses were run 10 times for each K value, and CV error was used to obtain the most probable K value for population structure analysis. ADMIXTURE plots were generated using 'Pophelper' in R (*Francis, 2017*). Genetic groups of accessions were assigned based on ancestry coefficient Q≥0.7, otherwise the accession was considered admixed. The population structure was also examined with PCA. The neighbor-joining phylogenetic tree was calculated using TASSEL (Trait Analysis by aSSociation, Evolution, and Linkage) software version 5.2.60 (*Bradbury et al., 2007*) and visualized using FigTree version 1.4.4 (http://tree.bio.ed.ac.uk/software/figtree/).

The relationships and gene flow among the four inferred genetic groups were further assessed by TreeMix version 1.12 (*Pickrell et al., 2012*) using 4396 LD-pruned SNPs with no missing data. The analysis was run for 0–3 migration events with *V. radiata* var. *sublobata* as an outgroup with a block size of 20 SNPs to account for the effects of LD between SNPs. We estimated one as the optimal number of migration events using the 'OptM' in R (*Fitak, 2021*). Bootstrap support for the resulting observed topology was obtained using 500 bootstrap replicates.

Nucleotide diversity (π) and genetic differentiation ($d_{xy}$ and $F_{ST}$) were estimated in 10 kb windows with pixy version 1.2.7.beta1 (*Korunes and Samuk, 2021*) using all 1,247,721 invariant and variant sites. LD decay for each genetic group was estimated based on 34,469 non-LD-pruned SNPs using PopLDdecay (*Zhang et al., 2019*). The curves were fitted by a LOESS function, and an LD decay plot was drawn using R.

To investigate the relation among inferred genetic groups, *f3* and *f4* statistics were computed based on filtered SNPs using ADMIXTOOLS version 7.0 (*Patterson et al., 2012*). The *f3* statistic

compares allele frequencies in two populations (A and B) and a target population C. In 'outgroup $f3$ statistic,' C is the outgroup, and positive values represent the shared genetic drift between A and B. In 'admixture $f3$ statistic,' negative values indicate that the C is admixed from A and B. For $f4$ statistics, $f4$(A, B; C, D) measures the shared genetic drift between B populations and C and D after their divergence from outgroup A. A positive value indicates that the B population shares more alleles with D, and a negative value indicates that the B population shares more alleles with C. We used two Mb as a unit of block-jackknife resampling to compute SEs. The Z-scores with absolute values greater than three are considered statistically significant.

To examine the role of geographic distance in shaping spatial genetic differentiation, Mantel tests with 1000 permutations were performed for each of the ADMIXTURE-inferred genetic groups (separately for the groups defined by Q≥0.7 or Q≥0.5) using 'ade4' in R. Pairwise genetic distance between accessions was estimated based on all sites while the great circle geographic distance was determined using 'fields' in R. In addition, the same analysis was conducted for southern and northern groups to examine if there was a south-north pattern of differentiation.

Based on the shape of the phylogenetic tree, we used fastsimcoal2 (*Excoffier et al., 2021*), which does not rely on whole-genome sequencing, to estimate the split time among genetic groups. Fifty accessions were randomly picked from each genetic group. Population size was allowed to change, and gene flow was allowed among populations. This analysis used all sites covered by the DArT tags (including monomorphic sites), and the mutation rate was set to $1 \times 10^{-8}$ which was within the range of mutation rates used in eudicots (*Barrera-Redondo et al., 2021*; *Zheng et al., 2022*). The models were run using unfolded site frequency spectrum using the major allele in the wild progenitor population (*V. radiata* var. *sublobata*) as the ancestral allele. The model was run independently 100 times, each with 100,000 simulations. After obtaining the run with the highest likelihood, we performed parametric bootstrapping 100 times to obtain the 75% CIs of each parameter based on the previous study of *Gutaker et al., 2020*.

## Ecological niche modeling

To understand whether the habitats of genetic groups are differentiated, 248 sampling sites (82 for EA, 45 for SEA, 49 for SA, and 72 for CA genetic groups), in combination with additional presence records obtained from the Global Biodiversity Information Facility (GBIF, https://www.gbif.org/), were used for the analysis. Using the longitude and latitude information, we extracted the Köppen climate zones (*Köppen, 2011*) using 'kgc' in R (*Bryant et al., 2017*). After excluding zones with less than 5 samples, the remaining 10 zones were grouped into 6 categories based on climate similarity: dry hot (BSh and BWh), dry cold (BSk and BWk), temperate dry summer (Csa), tropical savanna (Aw), continental (Dwb and Dfb), and temperate wet summer (Cfa and Cwa). The former three are relatively dry environments.

Climate layers comprising monthly minimum, maximum, mean temperature, precipitation, and 19 bioclimatic variables were downloaded from the WorldClim database version 1.4 (*Hijmans et al., 2005*). All climate layers available from WorldClim were created based on climate conditions recorded between 1960 and 1990 at a spatial resolution of 30 arc-seconds (approximately 1 km$^2$). To minimize redundancy and model overfitting, pairwise Pearson correlations between the 19 bioclimatic variables were calculated using ENMTools version 1.4.4 (*Warren et al., 2010*), excluding one of the two variables that has a correlation above 0.8. As a result, eight bioclimatic variables were used for all further analyses, including Bio1 (annual mean temperature), Bio2 (mean diurnal range), Bio3 (isothermality), Bio8 (mean temperature of wettest quarter), Bio12 (annual precipitation), Bio14 (precipitation of driest month), Bio15 (precipitation seasonality), and Bio19 (precipitation of coldest month). Bioclimatic variables were extracted for each occurrence point using 'raster' in R (*Hijmans, 2021*). PCA and MANOVA were conducted to examine whether there was a significant habitat difference among genetic groups. Ecological niche modeling (ENM) was performed using MAXENT version 3.3.1 (*Phillips et al., 2006*) to predict the geographic distribution of suitable habitats for cultivated mungbean. The ENM analysis was run with a random seed, a convergence threshold of 5000 and 10-fold CV. As a measure of the habitat overlaps of the four genetic groups, pairwise of Schoener's D was calculated using ENMTools. The value ranges from 0 (no niche overlap) to 1 (niche complete overlap). In addition, we carried out the same analyses using monthly temperature and precipitation from May, July, and September.

## Field evaluation

Among the 52 accessions used for laboratory experiments, phenotyping of 49 accessions was conducted at WorldVeg, Taiwan in 1984 and 2018 and at Crop Sciences Institute, National Agricultural Research Centre, Pakistan in 2015. The traits related to phenology (days to 50% flowering), reproduction (100 seed weight, pod length, pods per plant, 1000 seed weight, seeds yield per plant, and seeds per pod), and plant size (petiole length, plant height, plant height at flowering, plant height at maturity, primary leaf length, primary leaf width, terminal leaflet length, and terminal leaflet width) were included. Trait values were inverse normal transformed. The ANOVA was performed to test for inferred genetic groups differences for each trait using R software (version 4.1.0).

## Drought phenotyping

A total of 52 accessions with ancestry coefficients Q≥0.7 from three genetic groups (SEA, SA, and CA) were selected for experiments of seedling-stage drought response. The experiment was laid out in a completely randomized design with three replicates of each accession under two treatments (control/drought). The experiment was conducted in two independent batches, and the whole experiment included 624 plants (52 accessions × 2 treatments × 3 plants per treatment × 2 batches).

Mungbean seeds were surface-sterilized with 10% bleach for 10 min and rinsed with distilled water for three times. Seeds were treated with 70% ethanol for 5 min and washed three times in distilled water. The sterilized seeds were germinated on wet filter paper in petri dishes for 3 days. The experiment was conducted in a 740FLED-2D plant growth chamber (HiPoint, Taiwan) at a temperature of 25 ± 1°C and 12 hr of photoperiod (light ratios of red: green: blue 3: 1: 1) with light intensity 350 μmol m$^{-2}$s$^{-1}$ and relative humidity at 60 ± 5%. The seedlings were then transplanted to a hydroponic system with half-strength Hoagland nutrient solution (Phytotechnology Laboratory, USA) and were grown for 6 days before drought stress started. The nutrient solution was changed on alternate days, and the pH of the solution was adjusted to 6.0 with 1 M KOH or 1 M HCl.

For drought treatment, seedlings of mungbean were exposed to polyethylene glycol (PEG)-induced drought stress for 5 days. The solution of PEG6000 with an osmotic potential of –0.6 MPa was prepared by adding PEG6000 (Sigma-Aldrich, Germany) to the nutrient solution according to *Michel and Kaufmann, 1973*, and pH was also adjusted to 6.0. The seedlings grown with the nutrient solution under the same environmental conditions were considered as controls.

At the end of the experiment, plants were evaluated for SDW and root dry weight, measured on digital balance after oven-drying at 70°C for 48 hr. All traits were analyzed by mixed-model ANOVA with the treatment (control/drought) and the genetic group as fixed effects. The models included accessions as a random effect nested within genetic groups and a random effect of batches. Tukey's test was conducted to compare genetic groups. All statistics were performed using JMP v13.0.0 (SAS Institute, 2016).

## $Q_{ST}$-$F_{ST}$ comparisons

For each trait, quantitative trait divergence ($Q_{ST}$) was calculated separately with respect to each treatment. Our root and shoot weight experiment used a selfed-progeny design, using the self-fertilized seeds from each accession as replicates, as recommended for partially inbred species (*Goudet and Büchi, 2006*). For the selfed-progeny design of inbred species, (Equation 1) $Q_{ST} = V_B/(V_B+V_{Fam})$, where $V_B$ is the among-population variance component, and $V_{Fam}$ is the within-population among-family variance component (*Goudet and Büchi, 2006*). Variance components were estimated using a model with genetic groups, accessions nested within genetic groups, and batches as random factors. To accommodate the possibility that mungbean is not completely selfing, we also applied (Equation 2) $Q_{ST} = (1+f)V_B/([1+f]V_B +2 V_{AW})$ (*Goudet and Büchi, 2006*), where $f$ is the inbreeding coefficient (estimated by VCFtools as 0.8425), $V_B$ is the among-population variance component, and $V_{AW}$ is the additive genetic variance within genetic groups estimated by the kinship matrix using TASSEL software (*Bradbury et al., 2007*). The results and conclusions are similar to our previous version. The $F_{ST}$ was calculated only using accessions in the phenotyping experiment.

## Acknowledgements

We thank Chia-Yu Chen and Shang-Ying Tien for their assistance in sample preparation. C-RL was funded by 107–2923-B-002–004-MY3 and 110–2628-B-002–027 from the Ministry of Science and Technology, Taiwan. C-TT was funded by 107–2923-B-002–004-MY3 from the Ministry of Science and Technology, Taiwan. Y-PL was supported by 110–2313-B-125–001-MY3 from the Ministry of Science and Technology, Taiwan. RN and RS were funded by the Australian Center for International Agricultural Research (ACIAR) through the projects on International Mungbean Improvement Network (CIM-2014–079 and CROP-2019–144) and by the strategic long-term donors to the World Vegetable Center: Republic of China (Taiwan), UK aid from the UK government, United States Agency for International Development (USAID), Australian Centre for International Agricultural Research (ACIAR), Germany, Thailand, Philippines, Korea, and Japan. EBvW was supported by USDA Multistate Hatch NE2210 and USDA NIFA award 2022-67013-37120. MS was supported by the Ministry of Science and Higher Education of the Russian Federation as part of the World-class Research Center program: Advanced Digital Technologies (contract No. 075-15-2022-311 dated 20.04.2022). SN was supported by the Zumberge foundation.

## Additional information

### Funding

| Funder | Grant reference number | Author |
|---|---|---|
| Ministry of Science and Technology, Taiwan | 107-2923-B-002-004-MY3 | Chau-Ti Ting<br>Cheng-Ruei Lee |
| Ministry of Science and Technology, Taiwan | 110-2628-B-002-027 | Cheng-Ruei Lee |
| Australian Centre for International Agricultural Research | CROP-2019-144 | Ramakrishnan Madhavan Nair<br>Roland Schafleitner |
| Ministry of Science and Technology, Taiwan | 110-2313-B-125-001-MY3 | Ya-Ping Lin |
| Australian Centre for International Agricultural Research | CIM-2014-079 | Ramakrishnan Madhavan Nair<br>Roland Schafleitner |
| U.S. Department of Agriculture | Multistate Hatch NE2210 | Eric Bishop-von-Wettberg |
| Ministry of Science and Higher Education of the Russian Federation | 075-15-2022-311 | Maria Samsonova |
| USDA National Institute of Food and Agriculture | 2022-67013-37120 | Eric Bishop-von-Wettberg |
| Zumberge foundation | | Sergey Nuzhdin |
| Russian Science Foundation | 18-46-08001 | Eric Bishop-von-Wettberg<br>Marina Burlyaeva<br>Maria Samsonova<br>Margarita Vishnyakova |

The funders had no role in study design, data collection and interpretation, or the decision to submit the work for publication.

### Author contributions

Pei-Wen Ong, Data curation, Formal analysis, Validation, Investigation, Visualization, Writing – original draft, Writing – review and editing; Ya-Ping Lin, Hung-Wei Chen, Data curation, Formal analysis, Investigation, Writing – original draft; Cheng-Yu Lo, Data curation, Validation, Investigation; Marina Burlyaeva, Thomas Noble, Ramakrishnan Madhavan Nair, Roland Schafleitner, Margarita Vishnyakova, Eric Bishop-von-Wettberg, Maria Samsonova, Sergey Nuzhdin, Chau-Ti Ting, Data curation, Investigation,

Writing – original draft, Resources; Cheng-Ruei Lee, Conceptualization, Resources, Data curation, Formal analysis, Supervision, Funding acquisition, Investigation, Visualization, Methodology, Writing – original draft, Project administration, Writing – review and editing

### Author ORCIDs
Pei-Wen Ong  http://orcid.org/0000-0003-0494-4677
Ya-Ping Lin  http://orcid.org/0000-0002-9575-2007
Marina Burlyaeva  http://orcid.org/0000-0002-3708-2594
Thomas Noble  http://orcid.org/0000-0002-7731-5559
Ramakrishnan Madhavan Nair  http://orcid.org/0000-0002-2787-8396
Eric Bishop-von-Wettberg  http://orcid.org/0000-0002-2724-0317
Cheng-Ruei Lee  http://orcid.org/0000-0002-1913-9964

### Decision letter and Author response
Decision letter https://doi.org/10.7554/eLife.85725.sa1
Author response https://doi.org/10.7554/eLife.85725.sa2

## Additional files

### Supplementary files
• MDAR checklist

• Supplementary file 1. History of mungbean spread: genetic, environment, and traits data. (a) Mungbean accessions from Vavilov Institute (VIR) collection. (b) Outgroup $f3$ statistics among all possible combinations of genetic group pairs. (c) Admixture $f3$ statistics among all possible population trios. (d) Mantel tests for isolation by distance of inferred genetic group (Q≥0.5). (e) Description of bioclimatic variables used in ecological niche modeling. (f) Pearson's correlation coefficient between pairs of bioclimatic variables (denoted in lower triangle). (g) Comparison of bioclimatic variables among the four genetic groups analyzed with multivariate ANOVA (MANOVA). (h) Summary of ANOVA for bioclimatic variables. (i) Correlation between eight bioclimatic variables and climatic PC axes 1–4. (j) Comparison of summer growing season data including temperature and precipitation of May, July, and September among the four genetic groups analyzed with MANOVA. (k) ANOVA table for all evaluated field traits (phenology, reproduction, and size) as well as drought-related traits. (l) Mean of eight bioclimatic variables of the genetic groups

### Data availability
Sequences generated in this study are available under NCBI BioProject PRJNA809503. Accession names, GPS coordinates, and NCBI accession numbers of the Vavilov Institute accessions are available under Supplementary file 1a. Plant trait data are available at Dryad https://doi.org/10.5061/dryad.d7wm37q3h. Sequences and accession information of the World Vegetable Centre mini-core and the Australian Diversity Panel collections were obtained from the NCBI BioProject PRJNA645721 (*Breria et al., 2020*) and PRJNA963182 (*Noble et al., 2018*).

The following datasets were generated:

| Author(s) | Year | Dataset title | Dataset URL | Database and Identifier |
|---|---|---|---|---|
| Ong P, Lin Y, Chen H, Lo C, Noble T, Nair R, Schafleitner R, Vishnyakova M, Bishop-von-Wettberg E, Samsonova M, Nuzhdin S, Ting C, Lee C | 2023 | The climatic constrains of the historical global spread of mungbean | https://doi.org/10.5061/dryad.d7wm37q3h | Dryad Digital Repository, 10.5061/dryad.d7wm37q3h |
| Ong P, Lin Y, Chen H, Lo C, Noble T, Nair R, Schafleitner R, Vishnyakova M, Bishop-von-Wettberg E, Samsonova M, Nuzhdin S, Ting C, Lee C | 2023 | Vavilov Institute (VIR) mungbean collection - DArTseq | https://www.ncbi.nlm.nih.gov/bioproject/PRJNA809503 | NCBI BioProject, PRJNA809503 |

The following previously published datasets were used:

| Author(s) | Year | Dataset title | Dataset URL | Database and Identifier |
|---|---|---|---|---|
| Breria CM, Hsieh CH, Yen J-Y, Nair R, Lin C-Y, Huang S-M, Noble TJ, Schafleitner R | 2020 | World Vegetable Center Mini Core Collection - DartSeq | https://www.ncbi.nlm.nih.gov/bioproject/PRJNA645721 | NCBI BioProject, PRJNA645721 |
| Noble TJ, Tao Y, Mace ES, Williams B, Jordan DR, Douglas CA, Mundree SG | 2023 | Australian mungbean diversity panel collection - DArTseq | https://www.ncbi.nlm.nih.gov/bioproject/PRJNA963182 | NCBI BioProject, PRJNA963182 |

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

# Appendix 1

## Supplementary note

### Text analysis and translation of ancient Chinese texts regarding mungbean

### Qimin Yaoshu (齊民要術, about 544 AD)

Qimin Yaoshu, compiled by Sixie Jia (賈思勰), is one of the earliest and most complete agricultural sources in China, detailing agricultural techniques near the lower reaches of Yellow River at that era. This is the earliest record of mungbean in China, demonstrating mungbean has reached northern China at that time and is consistent with our estimates of population divergence time. The popularity of mungbean is demonstrated by it being mentioned multiple times under different contexts, most notably as a green manure:

「若糞不可得者，五六月中，概種菉豆，至七月，八月，犁掩殺之。如以糞糞田，則良美與糞不殊，又省功力。」

Translation: "Should feces be unavailable, during May and June one could grow mungbean. In July or August, one could plow mungbean plants into the soil. This is equivalent to using feces to manure the land. This is as good as using feces and saves efforts."

Notice that the months used in ancient China are slightly different from the Gregorian calendar.

### Xiangshan Yelu (湘山野錄, 1068–1077 AD)

Xiangshan Yelu was written by a monk, Wen-Ying (文瑩), recording anecdotes during that era. Its records about the Emperor Zhenzong of Song (宋真宗, 968–1022 AD) detailed the phenotypes of Indian mungbean at that time:

「真宗深念稼穡，聞占城稻耐旱，西天綠豆子多而粒大，各遣使以珍貨求其種。占城得種二十石，至今在處播之。西天中印土得菉豆種二石，不知今之菉豆是否？」

Translation: 'Zhenzong of Song deeply concerned about agriculture. He heard Champa rice being drought tolerant, and mungbean from India produce numerous and large seeds. Diplomats were sent to exchange the seeds with treasure. Twenty dans of Champa rice were obtained and propagated everywhere. Two dans of mungbean were obtained from India, but it is unclear whether the mungbean today descended from these.'

'Dan' (石) is a unit of volume in ancient China and is called 'Koku' in Japanese. The exact amount varied with time.

The texts provide us with two pieces of important information. First, mungbean from South Asia (likely also includes the SEA genetic groups if accessions near eastern India and Bangladesh were included) at that time had higher yield and larger seeds than native mungbean accessions in northern China, consistent with our results on trait divergence. Second, compared to the clear success of Champa rice in China, it was unclear whether those southern mungbean accessions had prospered in northern China, likely suggesting an unsuccessful introduction of southern high-yield and large-seeded accessions to the north.

### Tiangong Kaiwu (天工開物, 1637 AD)

Tiangong Kaiwu is a famous Chinese encyclopedia compiled by Song Yingxing (宋應星). While it mostly covers technologies at that time, a section about agricultural practices covers mungbean:

「綠豆必小暑方種，未及小暑而種，則其苗蔓延數尺，結莢甚稀。若過期至於處暑，則隨時開花結莢，顆粒亦少。」

Translation: 'Mungbean must be sown at or after Xiaoshu (Gregorian 7–8 July). Being sown before Xiaoshu, mungbean stems would spread for meters with few pods set. Being sown as late as Chushu (Gregorian 23–24 August), the plants would flower and set pods at any time, also with low yield.'

As a short-day plant, being sown too early when the days are too long, mungbean would have mostly vegetative growth. Being sown too late when the days are too short, flowering would be induced too quickly before sufficient vegetative development. In addition to our results that short-season accessions were favored in the north due to the requirement for drought escape, this source provides us with another support that mungbean could only be sown in a narrow time window due to daylength requirement. Given the autumn frost damage in the north, not being able to be sown earlier restricts the growing season length in the north, limiting the adoption of southern long-season accessions.

