## [Editor Report]

This is an important interdisciplinary effort, with compelling genetic evidence, that informs on the spread of an important crop. The work will be of broad interest to those studying the domestication and dissemination of cultivated plants.

---

## [Decision Letter]

**Decision letter after peer review:**

[Editors’ note: the authors submitted for reconsideration following the decision after peer review. What follows is the decision letter after the first round of review.]

Thank you for submitting the paper "The climatic constrains of the historical global spread of mungbean" for consideration by *eLife*.

First we would like to apologize for the length of time it took to reach this decision. We had hoped to get input from another archaeologist who had agreed to review but in the end did not submit a review. We had a rather long discussion of the work, plus several of the people involved were on vacation.

Your article has been reviewed by 3 peer reviewers, and the evaluation has been overseen by a Reviewing Editor and a Senior Editor. The following individual involved in the review of your submission has agreed to reveal their identity: Jeffrey Ross–Ibarra (Reviewer #1).

Comments to the Authors:

We are sorry to say that, after consultation with the reviewers, we have decided that this work will is not acceptable for publication in *eLife*.

The first two reviewers, both geneticists with experience in plant domestication, were excited about the topic and the interdisciplinary approach taken. But both raised a number of concerns about the methods and approaches used, which would require substantial additional work to address. The third reviewer, an archaeologist, raised real concerns about how well the paper has incorporated existing data into interpretation and analysis.

In the end, given the number of concerns raised on both the genetic and archaeological fronts, I'm sorry to say that we have decided that this work in its current form will not be considered further for publication by *eLife*. Having said this, there was broad interest in the work and the topic, and we would reconsider, albeit as a new submission, an extensively revised paper, which would likely look very different from the study at hand.

We are sorry about this outcome, but hope that the comments will be helpful for submission elsewhere.

*Reviewer #1 (Recommendations for the authors):*

The authors provide an ambitious interdisciplinary analysis of the post–domestication spread of mungbean. I believe the authors' general thesis is correct, and the various data they provide are all consistent with their model.

Nonetheless, there are some issues with analysis and interpretation, and in general, the data don't provide definitive evidence that allows claims about drought or the importance of the environment.

All of the evidence and analysis presented are consistent with the authors' proposed SA–SEA–EA–CA model. The f3 results are particularly convincing. I struggle, however, with how the data definitively show environment was more important than human movement. Some text shows, for example, that trading or demic movement between SA and CA was frequent would make clear that mungbean 'could' have moved SA–>CA via humans but didn't. Otherwise, could the SA–SEA–EA–CA model simply reflect historical movements/expansions of people in those directions/times?

On a similar note, the paper claims that drought was the most important factor limiting the mungbean movement. The phenotypic data certainly suggest adaptive differences in traits related to drought and drought differences in phenotype in an experimental setting, but additional evidence would be helpful to convince that drought is the key factor. The authors note the importance of daylength differences, for example – how do we know daylength wasn't the limiting factor (as it seems to have been in the northward spread of maize, for example)?

Methods Recommendations

– It's difficult in some places to know which SNP data set is used where. Are the LD–pruned SNPs used to estimate LD decay? The methods say there are 67K sites (including monomorphic) but there are 41K SNPs. This would imply that 2/3 of the sites are polymorphic. That can't be right so I must be misunderstanding something. Are these monomorphic sites also used for estimating nucleotide diversity (see the issue with VCFtools below)? For fastsimcoal, do you include sites where the derived allele is frequency=0 and where the derived allele is frequency 1?

– It would be good to modify the language of the MaxEnt modeling. While MaxEnt may show that the modern mungbean does not currently grow in conditions similar to those during the Holocene at a certain spot on the map, this is not the same as saying the plant could not have grown there. For example, if we were to model MaxEnt on modern CO2 concentrations, we would come to the conclusion that modern mungbean could not have grown anywhere during the Holocene. I don't think necessarily the methods need to be changed, but it would be good to change the language here to be less definitive.

– Although dominance is less of a concern in inbred species, I believe it is still incorrect to assume that using Vg is equivalent to Va in Qst. The epistatic variance could still be important, for example, and my brief internet search suggests mungbean isn't 100% selfing either. The authors have SNP data for all the individuals phenotyped, so it should be doable to estimate Va using kinship matrices and thus do a correct Qst analysis.

– VCFtools calculates nucleotide diversity assuming every bp in a window has been sequenced. This will lead to incorrect estimates of π and Fst. See https://pubmed.ncbi.nlm.nih.gov/33453139/ for details and one potential fix if you have a vcf with invariant sites, or https://github.com/RILAB/mop for another if you just have bams + variable vcf.

– an NJ tree (Figure 1E) is just a clustering algorithm and shouldn't be interpreted as providing information on the timing or order of evolutionary events. (lines 119–123)

– If the growing season is known and is what is most relevant for the crop (line 190), these should be used for most analyses or a justification for using annual data should be provided

– I can't find the total length of time plants were grown for the drought stress experiment. The text reads as if it were 9 days?

*Reviewer #2 (Recommendations for the authors):*

The study aims to understand the diffusion of the Mugbean population in Asia using genomic and phenotypic data. The authors used phylogeny, analysis of population structure, model–based inference, quantitative genetics, and niche modeling to understand the diffusion of Mugbean and proposed a scenario where diffusion is strongly constrained by climate, and partially by geographical.

I found the study very interesting with a mix of different methodologies from genomics to niche modeling and quantitative genetics. Few others studies used similar approaches but it is rather unique for the study of diffusion to combine these different approaches and bring up very interesting results. I have several comments on the methods so the results are better supported for the genetic part, for the phenotypic part, I am not sure the author will be able to support their claims, as the method they used might inflate variance in their estimations.

Main comments:

Phylogeny. Phylogeny is not really an appropriate method per see for intra–diversity study, phylogeny analysis could be used but the basic assumptions of the method are often inadequate, so the result should be put in context. A better method to do phylogeny–like inference is the TREEMIX approach based on the relationship of population integrating drift. I suggest the authors add this complementary analysis.

Model inference. Here, I would have liked a statistical comparison of the different scenarios. FastSimcoal allows for estimated probability of models/scenarios and it would be an independent validation to have such a comparison of models.

The inference of the model is dependent on mutation rate (unknown), notably on the time of divergence. The author used 10–8 but it is neither discussed nor justified.

Analysis of structure.

The author led aside from further studying individuals with ancestry lower than 70% in a given group. The analysis of structure could lead to such ancestry because of admixture or isolation by distance. One is secondary contact based on recent (or not) gene flow between genetic groups, the author is related to diffusion across the landscape. A large fraction of individuals in the PCA seems to me more likely related to isolation by distance (between SEA/EA, EA/CA). How this will impact the analysis of correlation by geographical distance if such individuals are not considered? The authors should perform the analysis of the Mantel test with all individuals to assess the impact of their choices.

Quantitative analysis of QST. The author used the genetic variance and not the additive variance. The reasoning behind that is not explained, neither the inflation created by this broad sense QST. Authors have concluded that the "extent to which comparisons between FST and broad–sense QST are appropriate remains unknown" (Pujol et al. 2008).

*Reviewer #3 (Recommendations for the authors):*

The paper aims to look at the domestication, post–domestication spread, and adaptation of mungbean (Vigna radiata) across Asia through the use of genetic data from landraces and accessions in see and genebanks.

The genetic aspects of this paper are a strength – there is a lot of work that has been done putting together a range of datasets allowing for inter–collection comparison and comparison of collections made by different institutes with their varying goals, sampling strategies, and dates of collecting. Mapping this diversity, think about how drift has occurred and why is something that needs to be done, especially in mungbean (and other tropical pulses of Asian/South Asian origin) as they are often overlooked in literature.

However, the main stated goal of the paper – to look at the domestication, post–domestication, and adaptations to climate change as this crop was moved around – is where it falls short. There is little engagement with the deep archaeological literature on both domestication as a process, post–domestication use and spread of mungbean (and other South Asian crops and those involved in the Silk Routes trade pathways), and the complexity of climate reconstructions and climate change over the stated period of interest/regions of interest. Works by Spengler and d'Alpoim Guedes for example are missed with regards to the Silk Routes debates, and literature by Fuller, Murphy given only short sentences as background for what is a very complex background regarding where and when mungbean is thought to have been domesticated. There is little reflection on the context of the two/three possible origins for mung (south, north, and west South Asia), how this interacts with the Southern Indian Neolithic and Indus regions, and how the changing cultural dynamics may have contributed to the processes of domestication, post–domestication change and the spread of different varieties. Without this background, it is hard to then move into discussing modern genetic data with a view to past patterns, for example with thinking about how climate may have affected change, given that the debates around the 4,2k event are extremely complex within these, let alone thinking pan–Asian and trying to link potentially 'drifted' genetic data today to these deep–time events. This comes across in the timescales given to the genetic data, as without the context of the where and when from archaeology, we see dates such as those given in Figure 2B that suggest domestication for South Asia moving to South East Asia at c.6kya (4000BC), which is when we still think they were under domestication within South Asia. The region is also not pinned down for where in South Asia these specific 'domesticated' mung are coming from to go to South East Asia, and the routes, yet arrows and big circles are added in Figure 2A. This shows the issue of not using that important context from archaeology.

A further issue arises from thinking about the climate data. By conflating vast areas (e.g.: South Asia, Central Asia, etc.), when applying climate modelling there has been an oversimplification, which makes any discussion of mung bean adapting to climate post domestication difficult to sustain. In line 184 for example, there is a suggestion on the role of the 6.2k event in Central Asia (putting aside the above issues of the dating of mung domestication in South Asia before it even reaches Central Asian regions already noted). While there are a few datasets as cited in the paper that show some impacts of wetter climates in some regions of Central Asia for a wetter 6,2k event this is by no means a universal impact, and regional data points are needed. We can see this when looking to the Indus and the impact of the 4.2k event as another example, again a point that needs refining in order to make such claims about mung domestication, let alone post–domestication adaptations.

Overall the thrust of the paper – domestication, post–domestication, and the spread of agriculture – are overshadowing what is actually a far more interesting point, hidden in lines 87–89: this data could be "used this resource to investigate the global history of mungbean after domestication […looking at the …] phenotypic characteristics for local adaptation to distinct environments." This perhaps is where the paper is most interesting, and reframing it in this context would be truly exciting, looking at the diversity of the crop, how it is now adapted to diverse environments, and what this might mean for long–term sustainability in cropping systems.

This paper sadly is losing some very interesting genetics data in complex and poorly explained mung history.

– The lack of engagement with archaeological data and the misunderstanding of the chronology of mung use in the past makes it very difficult to tally the results with the interpretations and discussions. This MAJOR point has been unpacked in more detail above and must be addressed in order to reduce the oversimplification of the background and remove the concerning premise that no one has done much work on ancient mung use (as stated up to l.77). While it has not had as much work as the cereals, there is still work being done on it, looking at its domestication regions, secondary domestication changes and spread across South Asia and then into different parts of Asia.

– Data seems to have been massaged to make it fit with the climate modelling in various regions (for example Figure 2B has mung arriving in Central Asia around 0.2k yet discussions of 6.2k climate events in l.184), and to also make mung seem to be spreading before the domestication event itself. More engagement with the archaeological discussions on mung domestication is needed and discussions of what domestication is as a process (there is a fundamental misunderstanding of the conscious/unconscious action outline in Larson et al. in l.47 – the way it is phrased implies deliberate choice to ensure change rather than recognizing the inherent unconscious and indirect action of human behavior and the entanglement of human–plant–environment interactions).

– Terms like cultivar and variety and landrace are used interchangeably. These must be defined in the paper. How they fit in with notions of domestication and post–domestication agricultural behavior must also be unpacked.

– "how the domesticated forms later expanded to a broader geographical area has also been detailed in several species, including maize (Matsuoka et al., 2002), rice (Huang et al., 2012), tomato (Razifard et al., 2020), chickpea (Varshney et al., 2021), and lettuce (Wei et al., 2021)." – these are unusual choices of case study to make a point as many (maize and rice as key examples) are not accepted as well defined and remain highly controversial. These are genetics papers, and demonstrate the lack of familiarity with the archaeological context of domestication. A quick glance at the literature around them will illustrate that they are poor choices of case studies to make this point as they too are highly controversial.

– "It is also unclear whether the expansion of most crops strictly follows the longitudinal axis of the continents (Diamond, 2005) or whether or why some are able to cross different climatic zones." – again poor knowledge of the archaeological context of these debates, and the reliance on Diamond is concerning as he is not an archaeologist. See works by Lister on barley and Lui on wheat as a good starting point.

– Debate on wild progenitor of mung bean needs to be explored. while sublobator is a likely candidate it should be explained in the paper that there are other possible options, and then why it was chosen here, with citations.

– The figures are difficult to use. This comes back to the conflation of space and time outlined in the public review. There are big circles on the maps covering the dots which I presume are either archaeological sites or accession points of the sampled beans(?! unclear), and then very odd choices of illustrating change over time. The figures are small and hard to see and require very long text in the figures to make them useable.

– Some aspects of basic geography have been overlooked to make climate the most important variable; l.166–7 "Given that geographic barrier might not be the most important factor". I find it hard to believe that both the Himalayas and major flood basins like the Brahmaputra would not be an issue, as would issues of day length when moving things north–south.

– In dealing with issues of climate change and adaptation some discussion of tolerance is needed. there must be a discussion (and a table perhaps) of the different watering, salination, temperature, etc. tolerance of the mung bean(s) under consideration to make the claims justified.

– Within the methods, the sampling strategy was hard to follow. This needs a more careful and clear description of decisions made: exactly where did the accessions come from geographically? how did their spread affect the dataset? Is there geographic clustering, did you compensate for that? how does the sampling potentially bias your data? a map would be useful.

– Fair and open protocols dictate that all methods must be stated: "Genomic DNA was extracted from a single plant per accession using QIAGEN Plant Mini DNA kit according to the manufacturer's instruction with minor modification." If you modified the protocol then you have to outline what you did so it can be reproducible and the data comparable.

– "Climate data for conditions between 1960–1990 were downloaded from the WORLDCLIM 1.4 database" – how was this dataset determined? why 30 years and not more? give citations to explain this decision, and look to other modelling efforts to check comparability.

– "19 bioclimatic variables" – what are these? why were they chosen? a table and explanation are needed.

– "excluding one of the two variables that have a correlation above 0.8 (Supplementary file 4)" – why? explain the reasoning for exclusion.

– Throughout the dataset has relied on "In total, our dataset contains more than one thousand accessions (1092) and covers worldwide diversity of mungbean representing a wide range of variation in seed colour" however at no point is there a discussion of whether these are modern variants of historic landraces and how this was assessed. This has a big impact on any discussion of "ancient" adaptations, and there must be a discussion of how you tested to see if the genetic changes you see are more recent or past changes and how the genetic clock was applied.

As noted in the public review, a far more interesting aspect than trying to tie into domestication/post–domestication and chronological vagaries are the points made in lines 87–89: this data could be "used this resource to investigate the global history of mungbean after domestication […looking at the …] phenotypic characteristics for local adaptation to distinct environments." Thinking about the value of this dataset for the preservation of diversity, and how diversity links to localised adaptations today and to sustainable cropping now is critical, and I suggest this could be the way to reframe things.

[Editors’ note: further revisions were suggested prior to acceptance, as described below.]

Thank you for resubmitting your work entitled "Environment as a limiting factor of the historical global spread of mungbean" for further consideration by *eLife*. Your revised article has been evaluated by Detlef Weigel as Senior and Reviewing Editor.

The manuscript has been improved but there are some remaining issues that need to be addressed, as outlined in the comments of Reviewer #1 below.

*Reviewer #1:*

I found this study to be one of the rare to combine genomic data, climate data, phenotypic data to decipher the diversity of adaptation of plants, and try to build up scenario of their diffusion. Previous recommendations were mainly answered, and I am personally satisfied with this new version of the manuscript. At this stage I recommend acceptance of the paper as soon as the concern is addressed.

Data availability:

I was not able to access the bioproject PRJN809503, is the data already available or not? Neither a biosample I try to access for a check.

Neither I was able to access DRYAD data.

The authors provide a link to fastq data but I could not find a fastq file link in Noble et al. or Breria et al. How did the authors merge the data? If they merged the data based on Table S1 or Noble et al. why did they find more SNPs than Noble et al. with a 10% missing rate? Since some authors are common between these studies a clear path to access to the whole dataset should be available to the community.

---

## [Author Response]

[Editors’ note: the authors resubmitted a revised version of the paper for consideration. What follows is the authors’ response to the first round of review.]

Reviewer #1 (Recommendations for the authors):The authors provide an ambitious interdisciplinary analysis of the post–domestication spread of mungbean. I believe the authors' general thesis is correct, and the various data they provide are all consistent with their model.1. Nonetheless, there are some issues with analysis and interpretation, and in general, the data don't provide definitive evidence that allows claims about drought or the importance of the environment.All of the evidence and analysis presented are consistent with the authors' proposed SA–SEA–EA–CA model. The f3 results are particularly convincing. I struggle, however, with how the data definitively show environment was more important than human movement. Some text shows, for example, that trading or demic movement between SA and CA was frequent would make clear that mungbean 'could' have moved SA–>CA via humans but didn't. Otherwise, could the SA–SEA–EA–CA model simply reflect historical movements/expansions of people in those directions/times?

Thank you. In the previous version we put most of the information in Discussion, and now we elaborated this point. Specifically, we emphasized studies that as early as about 4kya, Central and South Asia was connected through a complex exchange network linking the north of Hindu Kush, Iran, and the Indus River Civilization (Doumani Dupuy, 2016; Kohl, 2007; Kohl and Lyonnet, 2008; Lamberg-Karlovsky, 2002; Lombard, 2020; Lyonnet 2005). Later, there was archaeological evidences that diverse crops from West, South, and East Asia have been cultivated in northern Pakistan (Spengler *et al.* 2021), suggesting the prevalence of crop exchange. Human-mediated long distance dispersal of mungbean seeds also happened, as mungbean seeds have been found near the Red Sea Coast of Egypt during the Roman period (Van der Veen and Morales, 2015). Despite the prevalence of exchange, we note in Discussion: “Despite this, in Bronze-age archaeological sites north of Hindu Kush, cereals were frequently observed. Legumes (such as peas and lentils) were observed to a lesser extent, and South Asian crops were not commonly found (Jeong *et al.* 2019, Spengler *et al.* 2018). Interestingly, archaeologists suggested legume’s higher water requirement than cereals may be associated with this pattern, and pea and lentil’s role as winter crops in Southwest Asia may be associated with their earlier appearance in northern Central Asia than other legumes (Spengler *et al.* 2014).” Our study of genetics, climate, and plant traits therefore support conclusion from archaeological studies.

Doumani Dupuy PN 2016. Bronze Age Central Asia. The Oxford Handbook of Topics in Archaeology, New York: Oxford University Press.

Jeong *et al.* 2019. The genetic history of admixture across inner Eurasia. Nature Ecology and evolution 3:966-976

Kohl PL 2007. Entering a Sown World of Irrigation Agriculture – From the Steppes to Central Asia and Beyond: Processes of Movement, Assimilation, and Transformation into the “Civilized” World East of Sumer. The Making of Bronze Age Eurasia, Cambridge: Cambridge University Press pp. 182-243

Kohl PL and Lyonnet B 2008. By land and by sea: The circulation of materials and peoples, ca. 3500–1800 BC. In: Olijdam E and Spoor RH (eds.), Intercultural Relations between South and Southwest Asia. Studies in Commemoration of E.C.L. During Caspers (1934–1996), Oxford: Archaeopress pp. 29–42

Lamberg-Karlovsky CC 2002. Archaeology and language: The Indo-Iranians. Current Anthropology 43:63–88

Lombard P 2020.The Oxus civilization/BMAC and its interaction with the Arabian Gulf. A review of the evidences. In: Lyonnet B and Dubova NA (eds.), The world of the Oxus civilization, London: Routledge pp. 607-637

Lyonnet B 2005. Another possible interpretation of the Bactro-Margiana Culture (BMAC) of Central Asia: The tin trade. In: Jarriage C and Lefevre V (eds.), South Asia Archaeology 2001, Volume 1: Prehistory, Paris: Editions recherche sur les civilisations pp. 191-200

Spengler *et al.* 2014. Agriculturalists and pastoralists: Bronze Age economy of the Murghab alluvial fan, southern Central Asia. Vegetation History and Archaeobotany 23:805-820

Spengler *et al.* 2018. Arboreal crops on the medieval Silk Road: Archaeobotanical studies at Tashbulak. PLOS ONE 13(8):e0201409

Spengler *et al.* 2021. The southern Central Asian mountains as an ancient agricultural mixing zone: new archaeobotanical data from Barikot in Swat valley of Pakistan. Vegetation History and Archaeobotany 30:463-476

Van der Veen M and Morales J 2015. The Roman and Islamic spice trade: New archaeological evidence. Journal of Ethnopharmacology 167:54-63

2. On a similar note, the paper claims that drought was the most important factor limiting the mungbean movement. The phenotypic data certainly suggest adaptive differences in traits related to drought and drought differences in phenotype in an experimental setting, but additional evidence would be helpful to convince that drought is the key factor. The authors note the importance of daylength differences, for example – how do we know daylength wasn't the limiting factor (as it seems to have been in the northward spread of maize, for example)?

Thank you. The main thesis of this study is that environmental adaptation may be an important factor. We used drought (among many other environmental factors) as a test case and chose root/shoot traits and field phenology as supporting evidences. We wish to note that we do not claim drought to be the most important. We believe mungbean was influenced by the join effects of multiple environmental factors: In the north, there is a limited duration for sowing (due to daylength), and the soon-arriving fall frost (temperature) further limited the growing season. During this period the water availability is also low (drought). The rapid phenology therefore appears to be important for adaptation to the short growing season (caused by daylength and temperature) and declining water availability (drought). We have rephrased the point throughout the manuscript.

Methods Recommendations3. It's difficult in some places to know which SNP data set is used where. Are the LD–pruned SNPs used to estimate LD decay? The methods say there are 67K sites (including monomorphic) but there are 41K SNPs. This would imply that 2/3 of the sites are polymorphic. That can't be right so I must be misunderstanding something. Are these monomorphic sites also used for estimating nucleotide diversity (see the issue with VCFtools below)? For fastsimcoal, do you include sites where the derived allele is frequency=0 and where the derived allele is frequency 1?

We apologize for the error. After quality filtering, in total there are 1248K sites, among which 34K are SNPs. The nucleotide diversity, pairwise genetic distance, and all related analyses have been re-done using the estimates incorporating monomorphic sites. For fastsimcoal, we included monomorphic sites. The previous information (67K sites for fastsimcoal) was a typo. The LD decay was calculated using the SNP sites without LD-prune. We have revised the manuscript throughout.

4. It would be good to modify the language of the MaxEnt modeling. While MaxEnt may show that the modern mungbean does not currently grow in conditions similar to those during the Holocene at a certain spot on the map, this is not the same as saying the plant could not have grown there. For example, if we were to model MaxEnt on modern CO2 concentrations, we would come to the conclusion that modern mungbean could not have grown anywhere during the Holocene. I don't think necessarily the methods need to be changed, but it would be good to change the language here to be less definitive.

Thank you. After considering this and the comments of reviewer 3, we have removed the mid-Holocene niche modeling results. In this revision we focus on comparing the potential range overlap in current conditions.

5. Although dominance is less of a concern in inbred species, I believe it is still incorrect to assume that using Vg is equivalent to Va in Qst. The epistatic variance could still be important, for example, and my brief internet search suggests mungbean isn't 100% selfing either. The authors have SNP data for all the individuals phenotyped, so it should be doable to estimate Va using kinship matrices and thus do a correct Qst analysis.

Thank you. Goudet and Büchi (2006) showed that for partially inbred species, the selfed-progeny design (as in our study) is more suitable than the half-sib design. With selfed progeny, Goudet and Büchi (2006) suggested using Q_ST_ = V_B_ / ( V_B_ + V_Fam_ ) , where V_B_ is the among-group variance component and V_Fam_ is the family-level variance component within genetic groups. This is the method originally used in our study as well as in many *Arabidopsis thaliana* (Méndez-Vigo *et al.* 2013, Stenoien *et al.* 2004, Wieters *et al.* 2021) and *Medicago truncatula* (Bonnin *et al.* 1996) studies. To accommodate the possibility that mungbean is not completely selfing, we also applied the equation Q_ST_ = (1+*f*) V_B_/ ( (1+*f*) V_B_ + 2V_AW_ ) from Goudet and Büchi (2006), where *f* is the inbreeding coefficient (estimated by VCFtools as 0.8425), V_B_ is the among-population variance component, and V_AW_ is the additive genetic variance within populations estimated by the kinship matrix using TASSEL. The results and conclusions are shown in this revision and are similar to our previous version.

Bonnin *et al.* 1996. Genetic markers and quantitative genetic variation in *Medicago truncatula* (Leguminosae): a comparative analysis of population structure. Genetics 143:1795–1805

Goudet J and Buchi L 2006. The effects of dominance, regular inbreeding and sampling design on *Q_ST_*, an estimator of population differentiation for quantitative traits. Genetics 172(2):1337-1347

Mendez-Vigo *et al.* 2013. Among- and within-population variation in flowering time of Iberian *Arabidopsis thaliana* estimated in field and glasshouse conditions. New Phytologist 197(4):1332–1343

Stenoien *et al.* 2005. Genetic variability in natural populations of *Arabidopsis thaliana* in northern Europe. Molecular Ecology 14:137–148

Wieters *et al.* 2021. Polygenic adaptation of rosette growth in *Arabidopsis thaliana*. PLOS Genetics 17(1):e1008748

6. VCFtools calculates nucleotide diversity assuming every bp in a window has been sequenced. This will lead to incorrect estimates of π and Fst. See https://pubmed.ncbi.nlm.nih.gov/33453139/ for details and one potential fix if you have a vcf with invariant sites, or https://github.com/RILAB/mop for another if you just have bams + variable vcf.

We re-generated the vcf including invariant sites and excluded sites with high missing proportion. We used pixy (Korunes and Samuk, 2021) to calculate nucleotide diversity (π) and genetic differentiation (*F_ST_*) using all invariant sites. The results have been updated.

Korunes KL and Samuk K 2021. PIXY: Unbiased estimation of nucleotide diversity and divergence in the presence of missing data. Molecular Ecology Resources 21(4):1359-1368

7. An NJ tree (Figure 1E) is just a clustering algorithm and shouldn't be interpreted as providing information on the timing or order of evolutionary events. (lines 119–123)

Following the suggest from Reviewer 2, point 2, we used TREEMIX to build the population-level relationship. In addition, we used methods such as outgroup *f*3 tests and *f*4 tests to investigate the relationship among these genetic groups. All results are consistent with our previous results of (SA,(SEA,(EA,CA))).

8. If the growing season is known and is what is most relevant for the crop (line 190), these should be used for most analyses or a justification for using annual data should be provided

While mungbean could be grown in the summer/early fall for most of Asia, in the southern part of Asia mungbean are occasionally grown in winter/spring. Due to the discrepancy of growing season between the northern and southern parts of Asia, we therefore performed parallel analyses, one using annual data and the other using temperature and precipitation from May, July, and September as growing season data. Both results are presented in the revised manuscript.

9. I can't find the total length of time plants were grown for the drought stress experiment. The text reads as if it were 9 days?

We revised the Materials and methods section. The seedlings were grown in nutrient solution for 6 days before being subjected to drought stress. For drought treatment, seedlings of mungbean were exposed to polyethylene glycol (PEG)-induced drought stress for 5 days as illustrated in Author response image 1:

**Author response image 1. sa2fig1:** 

Reviewer #2 (Recommendations for the authors):The study aims to understand the diffusion of the Mugbean population in Asia using genomic and phenotypic data. The authors used phylogeny, analysis of population structure, model–based inference, quantitative genetics, and niche odelling to understand the diffusion of Mugbean and proposed a scenario where diffusion is strongly constrained by climate, and partially by geographical.1. I found the study very interesting with a mix of different methodologies from genomics to niche modeling and quantitative genetics. Few others studies used similar approaches but it is rather unique for the study of diffusion to combine these different approaches and bring up very interesting results. I have several comments on the methods so the results are better supported for the genetic part, for the phenotypic part, I am not sure the author will be able to support their claims, as the method they used might inflate variance in their estimations.

Thank you. For the phenotype part, in this revision we applied a method that specifically addressed additive genetic variance and the potential issue that mungbean is not fully inbred (Reviewer 1, point 5). The results are qualitatively similar and the conclusions are the same. Please refer to Reviewer 1, point 5 and Reviewer 2, point 5 for detail.

Main comments:2. Phylogeny. Phylogeny is not really an appropriate method per see for intra–diversity study, phylogeny analysis could be used but the basic assumptions of the method are often inadequate, so the result should be put in context. A better method to do phylogeny–like inference is the TREEMIX approach based on the relationship of population integrating drift. I suggest the authors add this complementary analysis.

Thank you. We have performed TREEMIX in Figure 2A. The relationship among these genetic groups was also supported by outgroup *f*3 statistics and *f*4 statistics.

3. Model inference. Here, I would have liked a statistical comparison of the different scenarios. FastSimcoal allows for estimated probability of models/scenarios and it would be an independent validation to have such a comparison of models.The inference of the model is dependent on mutation rate (unknown), notably on the time of divergence. The author used 10–8 but it is neither discussed nor justified.

Thank you. In addition to our previous results, in this revision we expanded the population structure inference section by incorporating and connecting independent analyses such as TREEMIX, *d_xy_*, *F_ST_*, outgroup *f*3 statistics, and *f*4 statistics. All results point to the same pattern that EA and CA are genetically closest, followed by SEA, and SA is the most diverged group. Regarding the tree shape, (SA,(SEA,(EA,CA))) has strong supports from many independent analyses. We therefore mainly use fastsimcoal2 not as a tree-shape-testing method but as a method to estimate population divergence time based on this tree shape, which has strong support from many independent methods.

We wish to note that the current tree shape is still consistent with multiple hypotheses of cultivar expansion (please refer to the manuscript for the “east hypothesis”, “north hypothesis”, and “northeast hypothesis”), and testing different tree shapes would not necessarily distinguish among these hypotheses. Instead of using fastsimcoal2 to test different tree shapes, we employed the evidence from π, linkage disequilibrium decay, and isolation by distance to test these hypotheses.

Upon checking the common mutation rates used in plant population genetics analyses, we found our original 1e^-8^ was within the range of mutation rates used in eudicots (literatures have used 1.43e^-9^ to 2.50e^-8^). We performed a parallel analyses using 2e^-8^, and the results are qualitatively similar, and we therefore retained the original results from 1e^-8^. The Author response table 1 shows the 75% confidence range of these time estimates (kga = thousand generations ago).

**Author response table 1. sa2table1:** 

Divergence	Mutation 1e^-8^	Mutation 2e^-8^
SA vs. (SEA,(EA,CA))	4.7 to 11.3 kga	5.6 to 11.3 kga
SEA vs. (EA,CA)	1.1 to 4.6 kga	1.3 to 5.4 kga
EA vs. CA	0.1 to 0.8 kga	0.1 to 0.5 kga

Analysis of structure.4. The author led aside from further studying individuals with ancestry lower than 70% in a given group. The analysis of structure could lead to such ancestry because of admixture or isolation by distance. One is secondary contact based on recent (or not) gene flow between genetic groups, the author is related to diffusion across the landscape. A large fraction of individuals in the PCA seems to me more likely related to isolation by distance (between SEA/EA, EA/CA). How this will impact the analysis of correlation by geographical distance if such individuals are not considered? The authors should perform the analysis of the Mantel test with all individuals to assess the impact of their choices.

Thanks for the suggestions. We have re-perform the Mantel test following the suggestion, and the results consistent with our previous IBD analysis.

5. Quantitative analysis of QST. The author used the genetic variance and not the additive variance. The reasoning behind that is not explained, neither the inflation created by this broad sense QST. Authors have concluded that the "extent to which comparisons between FST and broad–sense QST are appropriate remains unknown" (Pujol et al. 2008).

Thank you. Upon reading Pujol *et al.* (2008), it is unclear whether the comment pertains perdominately selfing species like mungbean. We therefore refer to Goudet and Büchi 2006 (https://pubmed.ncbi.nlm.nih.gov/16322514/), which has a complete treatment of the effect of inbredding and genetic variance estimates to *Q_ST_*. In addition to the suggestion from Goudet and Büchi 2006 (which is the same as our original approach), we follow the suggestions in Reviewer 1, point 5 and estimated the new *Q_ST_* using additive within-group genetic variance. The results are similar and the conclusions remain the same. Please also refer to Reviewer 1, point 5 for details.

Reviewer #3 (Recommendations for the authors):The paper aims to look at the domestication, post–domestication spread, and adaptation of mungbean (Vigna radiata) across Asia through the use of genetic data from landraces and accessions in see and genebanks.The genetic aspects of this paper are a strength – there is a lot of work that has been done putting together a range of datasets allowing for inter–collection comparison and comparison of collections made by different institutes with their varying goals, sampling strategies, and dates of collecting. Mapping this diversity, think about how drift has occurred and why is something that needs to be done, especially in mungbean (and other tropical pulses of Asian/South Asian origin) as they are often overlooked in literature.1. However, the main stated goal of the paper – to look at the domestication, post–domestication, and adaptations to climate change as this crop was moved around – is where it falls short. There is little engagement with the deep archaeological literature on both domestication as a process, post–domestication use and spread of mungbean (and other South Asian crops and those involved in the Silk Routes trade pathways), and the complexity of climate reconstructions and climate change over the stated period of interest/regions of interest. Works by Spengler and d'Alpoim Guedes for example are missed with regards to the Silk Routes debates, and literature by Fuller, Murphy given only short sentences as background for what is a very complex background regarding where and when mungbean is thought to have been domesticated. There is little reflection on the context of the two/three possible origins for mung (south, north, and west South Asia), how this interacts with the Southern Indian Neolithic and Indus regions, and how the changing cultural dynamics may have contributed to the processes of domestication, post–domestication change and the spread of different varieties. Without this background, it is hard to then move into discussing modern genetic data with a view to past patterns, for example with thinking about how climate may have affected change, given that the debates around the 4,2k event are extremely complex within these, let alone thinking pan–Asian and trying to link potentially 'drifted' genetic data today to these deep–time events. This comes across in the timescales given to the genetic data, as without the context of the where and when from archaeology, we see dates such as those given in Figure 2B that suggest domestication for South Asia moving to South East Asia at c.6kya (4000BC), which is when we still think they were under domestication within South Asia. The region is also not pinned down for where in South Asia these specific 'domesticated' mung are coming from to go to South East Asia, and the routes, yet arrows and big circles are added in Figure 2A. This shows the issue of not using that important context from archaeology.

We appreciate the valuable comments. In this revision we discussed previous archaeological findings in Introduction and, as suggested, use the archaeological findings as the basis of our genetic investigation. Using genetic data, we re-constructed the evolutionary relationship among these major genetic groups. We showed that CA and EA to be genetically closest, followed by SEA, and SA is the most diverged group. As stated in our revised Results section, this observation is actually still consistent with several hypotheses, and we used further population genetics evidences to distinguish among these hypotheses. While our data suggested the general trend of SA-SEA-EA-CA, our investigation is limited by the amount of samples with detailed geo-referenced information.

Indeed, archaeological evidences have shown multiple independent early cultivation of mungbean within South Asia. As the reviewer has pointed out, the present-day samples were ‘drifted’ from the ancient varieties. It is exactly because of this reason that we did not attempt to use the present-day samples’ detailed GPS coordinates to answer/discuss the issue about these independent origins of mungbean cultivation within South Asia. Due to the possibility of long-term seed exchange and replacement within South Asia, the samples collected in a specific location might not necessarily reflect the genetic information for ancient mungbean at the same locations. This is supported by our recent results (Lin *et al.* 2022) that present-day worldwide cultivars, despite their geographic origin, have the same haplotype in the promoter region of *VrMYB26a*, a candidate gene controlling pod shattering in several *Vigna* species. This is consistent with a single origin of the loss of pod shattering phenotype common to present-day cultivars and suggests that while early mungbean cultivations happened independently, eventually one form with the loss-of-pod-shattering phenotype (and *VrMYB26a* haplotype) dominated and replaced others. Answering where the "domestication allele" of *VrMYB26a* originated or deciphering the relationship between present-day germplasms and ancient cultivation origins within India would require information from more geo-referenced ancient DNA, which would be out of the scope of this present study. On the other hand, we recognize the climate heterogeneity within South Asia. We performed new climatic analyses separating the SA group into two major regions based on the Köppen climate classification (one more similar to Southeast Asia and the other more similar to Central Asia). The results remain similar.

We completely agree with the reviewer that modern samples are ‘drifted’ and do not necessarily reflect ancient conditions. It is exactly due to this reason that we did not attempt to pinpoint the detailed or specific route of the out-of-India event, either. The genetic data simply do not allow us to pinpoint, for example, whether mungbean expanded from South Asia to Southeast Asia / southern China through the land or maritime routes (Fuller *et al.* 2011; Castillo *et al.* 2017). In our revision, we emphasized these routes are both compatible with our hypothesis. We wish to emphasize that we do not put much emphasis on the exact route of mungbean expansion, nor do we think there is only one wave/one route of the out-of-India event. There might be multiple attempts to bring mungbean out of India (which could not be answered unless with ancient DNA), and our focus in this study is whether or when mungbean could be grown and became part of the local agriculture throughout Asia. As in many other population genetics studies, if the previous waves were replaced by the later-arriving germplasms, our data could not provide information for the previous waves. We agree using big circles and arrows are too rough for the more complicated history of mungbean spread. We have modified this figure and prevent specifying a specific route without genetic evidence.

Take human genetics as an example, while modern populations have drifted from ancient populations, studies using modern DNA revealed much of the important demographic history. Using genetic and phenotypic data from modern populations, scientists are able to decipher the unique adaptation in human history (for example, lactose intolerance and many other studies from researchers such as Rasmus Nielsen and Graham Coop). Our work has similar aims and approaches to these studies. Similarly, questions such as “how many independent out-of-Africa events have happened”, “whether each of these originated from different locations within Africa”, or “the specific route of how anatomically modern human reached East Asia” would await studies incorporating ancient DNA from archaeological studies. For mungbean, we look very much forward to the chance of investigating these samples.

About the estimated divergence time among genetic groups, we wish to note that the time estimates have high confidence intervals (previously we only used the 50% range, and Reviewer1 suggests reporting wider ranges). Further, the time estimated was in units of generations instead of years (we modified the manuscript to clarify this). The growing season in the northern part of Asia is short, and we reasonably assumed one generation per year. This might not fit the whole Asia, since the southern parts of Asian have longer growing season and may allow more than one generation per year. Assuming one generation per year (as we naively did in the previous version) overestimated the divergence time for the southern groups. This is the limitation of current methods in this field, and we have acknowledged this in the manuscript.

Castillo *et al.* 2016. Rice, beans and trade crops on the early maritime Silk Route in Southeast Asia. Antiquity 90(353):1255-1269

Fuller *et al.* 2011. Across the Indian Ocean: The prehistoric movement of plants and animals. Antiquity 85(328):544-558

Lin *et al.* 2022. Distinct selection signatures during domestication and improvement in crops: a tale of two genes in mungbean. bioRxiv

Stevens *et al.* 2016. Between China and South Asia: A Middle Asian corridor of crop dispersal and agricultural innovation in the Bronze Age. The Holocene 26(10):1541-1555

2. A further issue arises from thinking about the climate data. By conflating vast areas (e.g.: South Asia, Central Asia, etc.), when applying climate modelling there has been an oversimplification, which makes any discussion of mung bean adapting to climate post domestication difficult to sustain. In line 184 for example, there is a suggestion on the role of the 6.2k event in Central Asia (putting aside the above issues of the dating of mung domestication in South Asia before it even reaches Central Asian regions already noted). While there are a few datasets as cited in the paper that show some impacts of wetter climates in some regions of Central Asia for a wetter 6,2k event this is by no means a universal impact, and regional data points are needed. We can see this when looking to the Indus and the impact of the 4.2k event as another example, again a point that needs refining in order to make such claims about mung domestication, let alone post–domestication adaptations.

Our study started from population genetic analyses, which separated the samples into four genetic groups. These genetic groups were separated purely based on genetic data without any geographic information, and their names (SA, SEA, EA,CA) were given based on where most of the materials came from. Given this, each genetic group is a relatively homogeneous entity compared to worldwide genetic variation. We fully agree that each specific location has its own unique climatic conditions, but since the purpose of this study is to investigate what factors affect the divergence among these four genetic groups, we used each genetic group as a unit in climate analyses and aimed to identify which factors have larger among-group than within-group variation. We fully agree and recognize that the distribution ranges for some genetic groups have large environmental heterogeneity. For example, the EA group distributes from the Pacific Coast to Central Asia. This is why we separated the EA groups into the eastern and western halves to accommodate this issue. Another geographic region potentially with large environmental heterogeneity is South Asia. In this revision, we investigated the distribution of these samples based on Köppen climate classification. CA samples have relatively homogeneous distribution in a few Köppen zones with similar climatic characteristics, and so are SEA samples. Based on the patterns of SA and EA samples, we separated them into two groups each, resulting in a total of sis zones. We wish to emphasize that the major goal of this study is to investigate factors affecting the differentiation among these four genetic groups. Further subsetting the range into many small regions was not strongly supported by genetic evidence. Finally, niche modeling requires reasonable amount of samples to estimate the “tolerance environmental range” (Reviewer 3, point 1) of a group, and subsetting samples based on the unique climatic characteristics of each location is not suitable for this analysis.

Note that in climatic modeling, we obtained the present-day environments from the present-day distribution of these present-day materials to identify their suitable “niche space”. Using this information, we predicted “where would be the suitable environment for these genetic groups under different climatic conditions” given the niche space information of each genetic group. The existence of the projected distribution for the CA group at 6.2kya (in the previous supplement figure) was merely a method to compare the climatic difference between 6.2kya and today, and this figure does not mean we thought the CA group already existed in the Central Asia geographical region at 6.2kya. We did not attempt to make mungbean seem already widespread at that time (in response to Reviewer 3, point 5). The whole purpose of the analysis using 6.2k climate was to investigate whether the Central Asia geographical region may be a suitable habitat for plants from the SA group when the Central Asian climate might be slightly different from today. If so, that would reject our hypothesis. Due to the possibility of confusion, we removed this figure in the revision.

To project the suitable distribution range, this niche modeling method requires the GIS layer with data in every 1km geographical grid, which is available for very limited number of time points. While mid-Holocene is the only close-enough time point with bioclimatic variables being modeled (http://www.worldclim.com/past), we agree this is a poor choice. We also agree detailed regional data would better reflect the spatial heterogeneity of climatic conditions, but this distribution modeling approach requires climate data for all geographical grids. We therefore removed the mid-Holocene niche modeling result.

3. Overall the thrust of the paper – domestication, post–domestication, and the spread of agriculture – are overshadowing what is actually a far more interesting point, hidden in lines 87–89: this data could be "used this resource to investigate the global history of mungbean after domestication […looking at the …] phenotypic characteristics for local adaptation to distinct environments." This perhaps is where the paper is most interesting, and reframing it in this context would be truly exciting, looking at the diversity of the crop, how it is now adapted to diverse environments, and what this might mean for long–term sustainability in cropping systems.

Thank you. We fully agree understanding the genetics, unique phenotypic characteristics, and local adaptation of these germplasms is important, and we sincerely hope to use our results to contribute to the goal of long-term sustainability and crop improvement. This is the topic of other research projects in our group. On the other hand, we also agree with the two other reviewers that using the genetic data to contribute to understanding of mungbean cultivation expansion is interesting. As the potential importance of environmental adaptation in determining crop cultivation range has been suggested in several archaeological studies (Spengler *et al.* 2014 and Spengler *et al.* 2018), here we provide supports from the genetics perspective. We have modified the manuscript to reflect the archaeological context.

Spengler *et al.* 2014. Late Bronze Age agriculture at Tasbas in the Dzhungar Mountains of eastern Kazakhstan. Quaternary International 348:147-157

Spengler *et al.* 2018. The breadth of dietary economy in Bronze Age Central Asia: Case study from Adji Kui 1 in the Murghab region of Turkmenistan. Journal of Archaeological Science: Reports 22:372-381

This paper sadly is losing some very interesting genetics data in complex and poorly explained mung history.4. The lack of engagement with archaeological data and the misunderstanding of the chronology of mung use in the past makes it very difficult to tally the results with the interpretations and discussions. This MAJOR point has been unpacked in more detail above and must be addressed in order to reduce the oversimplification of the background and remove the concerning premise that no one has done much work on ancient mung use (as stated up to l.77). While it has not had as much work as the cereals, there is still work being done on it, looking at its domestication regions, secondary domestication changes and spread across South Asia and then into different parts of Asia.

Thank you. In our revision, we have devoted a paragraph in Introduction for the archaeological results.

5. Data seems to have been massaged to make it fit with the climate modelling in various regions (for example Figure 2B has mung arriving in Central Asia around 0.2k yet discussions of 6.2k climate events in l.184), and to also make mung seem to be spreading before the domestication event itself. More engagement with the archaeological discussions on mung domestication is needed and discussions of what domestication is as a process (there is a fundamental misunderstanding of the conscious/unconscious action outline in Larson et al. in l.47 – the way it is phrased implies deliberate choice to ensure change rather than recognizing the inherent unconscious and indirect action of human behavior and the entanglement of human–plant–environment interactions).

We wish to emphasize that we did not massage the data in any way. For the 6.2k climate issue, please refer to our response to Reviewer 3, point 2. We have modified the introduction to reflect the reviewer’s point about Larson *et al.* (2014).

6. Terms like cultivar and variety and landrace are used interchangeably. These must be defined in the paper. How they fit in with notions of domestication and post–domestication agricultural behavior must also be unpacked.

We apologise that we have not made it clearer about the definition of cultivar, variety and landrace. In our work, we defined landraces if accessions are collected from the countries traditionally cultivating them and locally adapted as well as lack of formal genetic improvement, ie. those conserved in the Vavilov Institute, many of which were collected in the early 20th century. In order to avoid confusion, we revised all sentences to clarify these terms in relevant places across the text.

7. "how the domesticated forms later expanded to a broader geographical area has also been detailed in several species, including maize (Matsuoka et al., 2002), rice (Huang et al., 2012), tomato (Razifard et al., 2020), chickpea (Varshney et al., 2021), and lettuce (Wei et al., 2021)." – these are unusual choices of case study to make a point as many (maize and rice as key examples) are not accepted as well defined and remain highly controversial. These are genetics papers, and demonstrate the lack of familiarity with the archaeological context of domestication. A quick glance at the literature around them will illustrate that they are poor choices of case studies to make this point as they too are highly controversial.

We have removed this citation and completely re-written this paragraph.

8. "It is also unclear whether the expansion of most crops strictly follows the longitudinal axis of the continents (Diamond, 2005) or whether or why some are able to cross different climatic zones." – again poor knowledge of the archaeological context of these debates, and the reliance on Diamond is concerning as he is not an archaeologist. See works by Lister on barley and Lui on wheat as a good starting point.

Thank you. We have removed the citation to Diamond and re-written this part to reflect the rich archaeological works.

9. Debate on wild progenitor of mung bean needs to be explored. while sublobator is a likely candidate it should be explained in the paper that there are other possible options, and then why it was chosen here, with citations.

Molecular phylogenetic based on internal transcribed spaces (ITS) sequences was carried with the aim to resolve the taxonomic contradiction in *Vigna* group (Goel *et al.* 2001). The constructed ITS-region-based trees revealed that *V. radiata* var. *sublobata* is closest to *V. radiata*.

Goel et al. 2001. Molecular evolution and phylogenetic implications of internal transcribed spaces sequences of nuclear ribosomal DNA in the *Phaseolus*-*Vigna* complex. Molecular Phylogenetic and Evolution 22(1):1-19

10. The figures are difficult to use. This comes back to the conflation of space and time outlined in the public review. There are big circles on the maps covering the dots which I presume are either archaeological sites or accession points of the sampled beans(?! unclear), and then very odd choices of illustrating change over time. The figures are small and hard to see and require very long text in the figures to make them useable.

Thank you. We responded the space and time issue in Reviewer 3, point 1 and Reviewer 3, point 2. In this revision we removed the circles and arrows and moved the figures to supplement. We still keep some supplement figures with circles and arrows, but these figures (Figure 2—figure supplement 2) should be merely treated as simple illustrations of potential hybridization scenarios instead of detailed routes.

11. Some aspects of basic geography have been overlooked to make climate the most important variable; l.166–7 "Given that geographic barrier might not be the most important factor". I find it hard to believe that both the Himalayas and major flood basins like the Brahmaputra would not be an issue, as would issues of day length when moving things north–south.

Thank you. Since pods of mungbean cultivars do not shatter naturally, they have lost the natural ability to disperse seeds. We therefore focus on human activities when we discuss the ability to disperse. We are fully aware that it is difficult for human to cut through the Himalayas directly and we did not claim the Himalayas are not a barrier. Previously we put our argument mainly in Discussion, and here we elaborated part of the points in Results. Specifically, we emphasized that South and Central Asia was connected by a complex exchange network among north of Hindu Kush, Iran, and the Indus Valley. Between South and Southeast Asia, we did not assign a specific route, either. It is equally likely that mungbean spread through land or maritime routes. As emphasized in Reviewer 3, point 1, we focus on the relationship among the genetic groups instead of designating the exact routes. We have modified the manuscript reflect this.

In this study, “geography” mostly refers to landscapes hindering human activity and the movement (such as the Himalayas and the Brahmaputra). We regard daylength (as well as temperature, growing season length, and water availability) as a environmental factor that might limit the successful cultivation of mungbean in a new environment even though human had successfully transported mungbean there as seeds. We have rephrased this part to reflect the reviewer’s comment.

12. In dealing with issues of climate change and adaptation some discussion of tolerance is needed. there must be a discussion (and a table perhaps) of the different watering, salination, temperature, etc. tolerance of the mung bean(s) under consideration to make the claims justified.

Thanks for your comment. We have now revised the discussion comprehensively, mentioning that the precipitation in Central Asia is greatly below the lower limit of standard optimal conditions to grow mungbean in the south. To the best of our knowledge, this is the first study to systematically compare the phenotypic characteristics of different genetic groups in mungbean.

13. Within the methods, the sampling strategy was hard to follow. This needs a more careful and clear description of decisions made: exactly where did the accessions come from geographically? how did their spread affect the dataset? Is there geographic clustering, did you compensate for that? how does the sampling potentially bias your data? a map would be useful.

Thank you. We have clarified in Materials and methods that while all samples have the information of their countries of origin, only a subset of samples have the detailed longitude and latitude coordinates. These samples were used in the detailed analyses connecting genetic and location information (the isolation by distance analyses). The genetic groups (SA, SEA, EA, CA) were named from the geographic region where most of their members originated from, by considering both country and geographic coordinate information. Only samples from Asia were used. In this revision, we also put the accession location maps in supplement figures. About how the distribution of sample might affect our conclusion, we added the following note in Discussion:

“We recognize that not all samples have available spatial data, and we do not have samples from some parts of Asia. For example, while most samples of the SEA group were collected from Taiwan, Thailand, and Philippines, we do not have many samples from the supposed contact zone between SA and SEA (Bangladesh and Myanmar) or between SEA and EA (southern China). If more samples were available from these contact zones, the modeled niche space between SA and SEA and between SEA and EA would be even more similar than the current estimate, strengthening our hypothesis that niche similarity might facilitate the cultivation expansion. On the other hand, clear niche differentiation between SA and CA was evident despite the dense sampling near their contact zone.”

14. FAIR and open protocols dictate that all methods must be stated: "Genomic DNA was extracted from a single plant per accession using QIAGEN Plant Mini DNA kit according to the manufacturer's instruction with minor modification." If you modified the protocol then you HAVE to outline what you did so it can be reproducible and the data comparable.

The modifications made during DNA extraction were added.

15. "Climate data for conditions between 1960–1990 were downloaded from the WORLDCLIM 1.4 database" – how was this dataset determined? why 30 years and not more? give citations to explain this decision, and look to other modelling efforts to check comparability.

Using the WorldClim database has now become a common practice in the field of environmental niche modeling. The WorldClim database used meteorological station data collected during 1960-1990 to extrapolate worldwide climatic conditions in about 1km^2^ (30s) resolution. Therefore, we directly used the available climate layers which were created based on climate conditions recorded between 1960 and 1990 to predict the geographic distribution of suitable habitats for cultivated mungbean. We have rephrased this part.

16. "19 bioclimatic variables" – what are these? why were they chosen? a table and explanation are needed.

Explanation of each bioclimatic variables used in this study are available on WorldClim website (https://www.worldclim.org/data/bioclim.html#google_vignette) and we have added a table in supplementary file 5. These are often used in species distribution modelling and any related ecological modelling techniques.

17. "excluding one of the two variables that have a correlation above 0.8 (Supplementary file 4)" – why? explain the reasoning for exclusion.

The purpose of removing the highly correlated variables was to minimise the effect of multicollinearity, which could result in the overfitting of the model. This is the standard method in the field of ecological niche modelling and studies related to the environment variables (Coulibaly *et al.* 2022; Gao *et al.* 2021 and Zhao *et al.* 2019).

Coulibaly *et al.* 2022. Coupling genetic structure analysis and ecological-niche modeling in Kersting’s groundnut in West Africa. Scientific Reports 12:5590

Gao *et al.* 2021. Combined genotype and phenotype analyses reveal patterns of genomic adaptation to local environments in the subtropical oak *Quercus acutissima*. Journal of Systematics and Evolution 59(3):541-556

Zhao *et al.* 2019. Resequencing 545 ginkgo genomes across the world reveals the evolutionary history of the living fossil. Nature Communications 10:4201

18. Throughout the dataset has relied on "In total, our dataset contains more than one thousand accessions (1092) and covers worldwide diversity of mungbean representing a wide range of variation in seed colour" however at no point is there a discussion of whether these are modern variants of historic landraces and how this was assessed. This has a big impact on any discussion of "ancient" adaptations, and there must be a discussion of how you tested to see if the genetic changes you see are more recent or past changes and how the genetic clock was applied.

Thank you. Our materials were directly obtained from worldwide stock centers, which house germplasms mostly collected during the 20th century. The missions of these stock centers are the collection and conservation of local landraces instead of developing or breeding novel varieties, and therefore our materials have limited numbers of “breeder lines” from recent artificial crosses. Since we focused on the comparison among the four genetic groups, recent admixture within the same group would have limited impact on the among-group pattern, and cross-group admixed accessions were excluded from most analyses. Similarly, despite generations of admixture, human geneticists could use contemporary human DNA and phenotypes to infer the divergence time and adaptation among major worldwide populations, as long as recently admixed individuals were excluded. In addition, as we noted in Introduction, the majority of accessions with detailed collection site information (from the Vavilov Institute) were collected during the early 20th century, before large-scale seed exchanges among worldwide stock centers. As noted by Jones *et al.* (2008), "The efforts of collectors such as N. I. Vavilov have prevented the extinction of some of these local ecotypes.", our use of the Vavilov institute collection is therefore important.

We, however, acknowledge that genetic results from these materials collected during the 20th century may not reflect specific changes in specific time points in the past. This may be solved by ancient DNA analyses and is out of the scope of this current work. About divergence time, the fastsimcoal2 analysis was specifically used to address this issue while allowing potential gene flow among groups. Finally, we acknowledge that it may be difficult to estimate when the adaptive phenotypes originated. This may require specific modelling on whole-genome sequencing data and the identification of candidate genes, which would be out of the scope of the current study. Therefore, we used the descriptions in ancient Chinese texts about plant phenotypes as a supporting evidence to show some of these phenotypic characteristics appeared long time ago.

Jones *et al.* 2008. Approaches and constraints of using existing landrace and extant plant material to understand agricultural spread in prehistory. Plant Genetic Resources 6(2):98-112

19. As noted in the public review, a far more interesting aspect than trying to tie into domestication/post–domestication and chronological vagaries are the points made in lines 87–89: this data could be "used this resource to investigate the global history of mungbean after domestication […looking at the …] phenotypic characteristics for local adaptation to distinct environments." Thinking about the value of this dataset for the preservation of diversity, and how diversity links to localised adaptations today and to sustainable cropping now is critical, and I suggest this could be the way to reframe things.

Thank you. Please see our response to Reviewer 3, point 3.

[Editors’ note: what follows is the authors’ response to the second round of review.]

The manuscript has been improved but there are some remaining issues that need to be addressed, as outlined in the comments of Reviewer #1 below.Reviewer #1:I found this study to be one of the rare to combine genomic data, climate data, phenotypic data to decipher the diversity of adaptation of plants, and try to build up scenario of their diffusion. Previous recommendations were mainly answered, and I am personally satisfied with this new version of the manuscript. At this stage I recommend acceptance of the paper as soon as the concern is addressed.Data availability:I was not able to access the bioproject PRJN809503, is the data already available or not? Neither a biosample I try to access for a check.Neither I was able to access DRYAD data.

Thank you for your valuable comments and suggestions. Regarding this comment specifically, all the sequences generated from this study was already deposited to NCBI (Bioproject PRJNA809503). As for phenotyping measurement data was also deposited to DRYAR. We can provide reviewer links (as in Materials Design Analysis Reporting, MDAR) as below:

NCBI:

https://dataview.ncbi.nlm.nih.gov/object/PRJNA809503?reviewer=g4jqbn30fhpacs7di7vgj2b3ok

DRYAD:

https://datadryad.org/stash/share/b5TO7fUNguxu0hMp6zx1cQZkBrCKzDmznCPvzf_CqEs

We will ensure the whole dataset is available to the public before publication online.

The authors provide a link to fastq data but I could not find a fastq file link in Noble et al. or Breria et al. How did the authors merge the data? If they merged the data based on Table S1 or Noble et al. why did they find more SNPs than Noble et al. with a 10% missing rate? Since some authors are common between these studies a clear path to access to the whole dataset should be available to the community.

We apologize for not describing this clearly in the manuscript. We re-preformed the SNP calling step for all accessions from Vavilov Institute (VIR), Australian Diversity Panel (ADP) (Noble et al., 2018) and the World Vegetable Center (WorldVeg) mini-core (Breria et al., 2020). Additionally, we included BioProject information to the published dataset by Noble et al. (2018) and Breria et al. (2020) under Data Availability (Line 643). The detailed information of the whole dataset as listed in Author response table 2:

**Author response table 2. sa2table2:** 

Title	Reference	BioProject	Status
Australian mungbean diversity panel collection – DArTseq	Noble et al., 2018	PRJNA963182	Available
World Vegetable Center Mini Core Collection – DartSeq	Breria et al., 2020	PRJNA645721	Available
Vavilov Institute (VIR) mungbean collection – DArTseq		PRJNA809503	Release immediately following publication